# Integrated multiplexed assays of variant effect reveal determinants of catechol-*O*-methyltransferase gene expression

Ian Hoskins [ID], Shilpa Rao, Charisma Tante & Can Cenik [ID] ✉

## Abstract

Multiplexed assays of variant effect are powerful methods to profile the consequences of rare variants on gene expression and organismal fitness. Yet, few studies have integrated several multiplexed assays to map variant effects on gene expression in coding sequences. Here, we pioneered a multiplexed assay based on polysome profiling to measure variant effects on translation at scale, uncovering single-nucleotide variants that increase or decrease ribosome load. By combining high-throughput ribosome load data with multiplexed mRNA and protein abundance readouts, we mapped the *cis*-regulatory landscape of thousands of catechol-*O*-methyltransferase (*COMT*) variants from RNA to protein and found numerous coding variants that alter *COMT* expression. Finally, we trained machine learning models to map signatures of variant effects on *COMT* gene expression and uncovered both directional and divergent impacts across expression layers. Our analyses reveal expression phenotypes for thousands of variants in *COMT* and highlight variant effects on both single and multiple layers of expression. Our findings prompt future studies that integrate several multiplexed assays for the readout of gene expression.

**Keywords** COMT; MAVE; Mutagenesis; Ribosome Load; RNA Secondary Structure
**Subject Categories** Chromatin, Transcription & Genomics; RNA Biology

## Introduction

Genome-wide association studies have identified regulatory variants that impact RNA expression, transcript isoform usage, and translation (Lappalainen et al, 2013; Li et al, 2013; Battle et al, 2015; Cenik et al, 2015; GTEx Consortium, 2020); however, these methods have been limited to surveying common population variants. Consequently, interpreting the impact of rare genetic variation on gene expression remains a major challenge. Multiplexed assays of variant effect (MAVEs) employ deep sequencing to profile the phenotypic effects of thousands of genetic variants in a single target gene at once (Esposito et al, 2019). MAVEs have been developed to assay specific phenotypic readouts such as protein activity (Starita et al, 2015; Majithia et al, 2016; Suiter et al, 2020; Chiasson et al, 2020), protein-protein interactions (Fowler et al, 2010; Starr et al, 2020; Faure et al, 2022), mRNA abundance (Findlay et al, 2018; Kircher et al, 2019), protein abundance (Matreyek et al, 2018; Suiter et al, 2020; Chiasson et al, 2020), and organismal fitness (Sun et al, 2016; Weile et al, 2017).

Even though variant effects on several layers of gene expression have been assayed, no MAVE has been developed to measure coding variant effects on translation at scale in native coding sequences. Past efforts have focused on quantifying the ribosome occupancy of reporters with randomized 5′ UTRs (Sample et al, 2019) or unnatural sequence elements such as poly-charged peptides (Tesina et al, 2020; Burke et al, 2022). In this study, we developed a high-throughput assay for translation based on polysome profiling, which measures the number of ribosomes on mRNAs via sucrose gradient fractionation and RNA-sequencing. Utilizing our established software for MAVE analysis (Hoskins et al, 2023), we profiled variant effects on translation at single-nucleotide resolution, and integrated these translation measurements with MAVEs measuring mRNA and protein abundance for catechol-*O*-methyltransferase (*COMT*), an enzyme of major biomedical significance.

COMT catabolizes dopamine, epinephrine, norepinephrine, and catechol estrogens. Several common population SNPs of *COMT* were significant in genome-wide association studies of addictive behavior, cognition, and mood and neurological disorders (Egan et al, 2001; Shifman et al, 2002; Bray et al, 2003; Meyer-Lindenberg et al, 2006; Hendershot et al, 2012; Guillot et al, 2015; Lin et al, 2017). Drugs targeting COMT are used in the treatment of neurological disorders such as Parkinson's disease (Bialecka et al, 2008; Ruottinen and Rinne, 1998). Further, COMT breaks down catechol estrogens which can be genotoxic and may contribute to cancer (Dawling et al, 2001). In prostate and breast cancer cell lines, overexpression of *COMT* inhibited cancer progression by limiting cell invasion and enhancing apoptosis (Hashimoto et al, 2021; Janacova et al, 2023). In addition, 2-methoxyestradiol, a product of COMT catabolism of 2-hydroxyestradiol, inhibits cancer progression (Fotsis et al, 1994; Klauber et al, 1997; Mukhopadhyay and Roth, 1998), suggesting *COMT* plays a suppressive role in cancer.

Department of Molecular Biosciences, University of Texas at Austin, Austin, TX 78712, USA. ✉E-mail: ccenik@austin.utexas.edu

While variant effects on COMT protein function have been described for the common population variants rs4680, rs4818, and rs4633 (Chen et al, 2004; Nackley et al, 2006), and a noncoding variant in exon 1 (NM_001135162.2) (Lim et al, 2021), there is little annotation of rare variant effects on COMT gene expression. For example, expression phenotypes are not mapped for most COMT variants annotated with "drug response" in the ClinVar database (Landrum et al, 2016).

COMT is produced as both a membrane (MB-) and soluble (S-COMT) protein isoform, with alternative translation initiation producing the S-COMT isoform. It was previously suggested that common coding variants near the translation initiation site (TIS) of S-COMT influence protein abundance and enzyme activity, possibly through alteration of mRNA secondary structure (Nackley et al, 2006; Tsao et al, 2011). Thus, we sought to annotate rare COMT variants in the same coding region and test the possibility that coding variants influence expression through alteration of RNA secondary structure.

## Results

### Predicted RNA structure in the COMT coding region is supported by experimental measurements

Prior studies used computational predictions to nominate a regulatory role of mRNA secondary structure in the early coding region of COMT (Nackley et al, 2006; Tsao et al, 2011). In line with previous studies, we found that the 5′ UTR of COMT was folded with a modest probability, and the first coding exon, which coincides with clinically relevant variants (Landrum et al, 2016), was predicted to be highly structured (exons 2 and 3 of NM_000754.4, Fig. 1A, "Methods").

Since an increasing number of experimental methods have been deployed to assess RNA secondary structure, we next analyzed RNA probing data from the RASP database (Li et al, 2021) to assess experimental evidence for structure. Dimethyl sulfate (DMS) profiling methods yielded signal in COMT coding and untranslated regions. DMS-seq (Rouskin et al, 2014), which is based on RT termination and mapping of truncated 3′ ends, indicated most regions were relatively unstructured in K562 cells; however, DMS-MaPseq (Zubradt et al, 2017), a mutational profiling method that relies on DMS scar readthrough and mismatch enumeration, indicated strong secondary structure across most regions of exons 2 and 3 in HEK293T cells (Fig. 1B). Both methods indicated higher structure just downstream of the TIS for the MB-COMT isoform, which overlaps the predicted stem structures from mFold. In addition, DIM-2P-seq (Wu and Bartel, 2017), which enriches structures near the 3′ end, indicated potential structure in the 3′ UTR (Fig. EV1A); however, the pattern of signal in this region may also indicate expression of alternative transcript isoforms (Appendix Note).

Our analysis of predicted and experimental secondary structure data indicates structure in the 5′ UTR (exon 2), coding exon 3, and potentially the 3′ UTR (exon 6). Based on these results, we hypothesized disruption of mRNA structure and possibly alternative translation initiation in exon 3 may cause expression changes identifiable at multiple levels of gene expression (mRNA abundance, ribosome load, and protein abundance) (Nackley et al, 2006; Tsao et al, 2011; Lim et al, 2021).

### Characterization of COMT uORFs and common population SNPs

To dissect regulatory elements impacting COMT expression, we generated a HEK293T cell line that stably expressed the conserved 5′ UTR exon and MB-COMT coding sequence with a N-terminal Flag tag and a C-terminal fluorescent tag from a landing pad locus (Matreyek et al, 2020; "Methods"). To better characterize the translation pattern of our transgene, we conducted ribosome profiling experiments and found evidence for translation initiation, not only at the MB-COMT start codon, but also at two upstream open reading frames (uORFs) in the 5′ UTR (Fig. 1C). Both uORFs (hereafter uORF A and B), showed high ribosome footprint signal at CUG start codons in previous translation initiation site (TIS)-sequencing data in multiple cell types (Michel et al, 2014; Fig. EV1B).

We characterized the potential effect of uORF translation on gene expression using several single-nucleotide variants and deletions (Figs. EV1C–E and EV2, Appendix Note). Our analysis revealed potential regulation of MB-COMT expression by uORF B. We also characterized the effect of common population variants in the coding region on COMT expression (Fig. EV3, Appendix Note). Altogether, we noted possible alteration of COMT expression by noncoding variants but not common coding variants, although these effects may be due to our transgene design (see "Discussion"). To rigorously annotate the large space of possible variant effects in the coding region of COMT, we proceeded by characterizing variant effects on RNA abundance, ribosome load, and protein abundance after mutagenizing two COMT regions spanning the locations of common population SNPs.

### Design of multiplexed assays for COMT variant effect analysis

To scale our measurements of variant effect, we had mutagenesis libraries synthesized to generate all alternate silent and missense codons, and one nonsense codon, at each of 58 amino acid positions in COMT (Fig. 2A, "Methods"). We focused our mapping efforts on two regions spanning COMT variants of clinical significance (Landrum et al, 2016; Fig. EV3A). The first mutagenesis region (ROI 1) overlaps the predicted mRNA stem structure in exon 3, and the second region (ROI 2) covers the SNPs in exon 4 that were reported to modulate protein abundance and enzyme activity (rs4680, rs4818) (Nackley et al, 2006, 2009). We opted to target these regions instead of catalytic regions of COMT to interrogate variant effects tied to RNA folding and codon usage in the early coding sequence of COMT. Each library exhibited high uniformity of representation of intended alterations (Fig. EV4A), and multi-nucleotide polymorphisms (MNPs) were at globally lower frequencies compared to SNPs (Fig. EV4B).

After stable expression of each COMT library in four replicates, we sequenced the mutagenized regions using genomic DNA, total RNA, and polysomal RNA from four pooled fractions after sucrose gradient fractionation ("Methods"). To measure mRNA and protein abundance by an orthogonal method, we used flow cytometry to bin cells into one of four populations based on moxGFP (protein) and mCherry (mRNA proxy) fluorescence. In total, we generated over 120 sequencing libraries for all gene expression readouts.

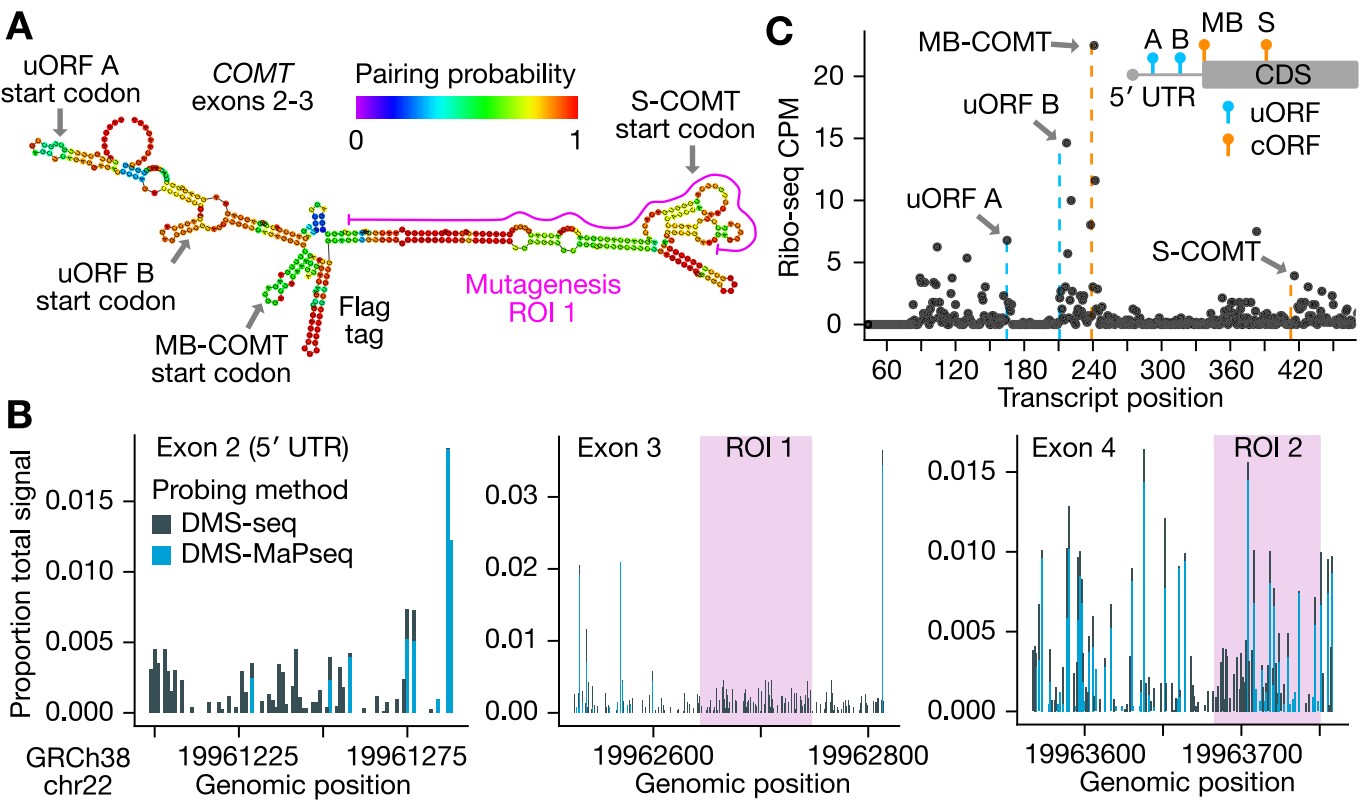

**Figure 1.  *COMT* RNA secondary structure and uORFs.**

(A) Predicted *COMT* mRNA structure. RNAfold was used to predict RNA secondary structure in *COMT* exons 2 and 3 (NM_000754.4, including a Flag tag). Highlighted are the locations of the uORF and canonical membrane and soluble isoform translation initiation sites, and the first mutagenesis region (ROI 1). (B) *COMT* RNA probing data. Dimethylsulfate (DMS) probing data available from the RASP database (Li et al, 2021) is shown. Low or lack of signal in DMS probing data indicates more RNA structure. DMS-seq was carried out in K562 cells (Rouskin et al, 2014), and DMS-MaPseq data in HEK293T cells (Zubradt et al, 2017). DMS-MaPseq showed little to no signal in exons 2 and 3 relative to DMS-seq. (C) Translation initiation in the 5′ UTR of COMT. Ribosome protected fragment (RPF) 5′ ends of one wild-type biological replicate are plotted as counts per million reads (CPM) following unique molecular identifier based PCR deduplication. Vertical dotted lines mark the expected 5′ end of RPFs when the P-site is positioned at the start codons of the candidate ORFs. Top right inset is a schematic of the ORF locations. ROI region of interest, UTR untranslated region, uORF upstream open reading frame, cORF canonical open reading frame.

After excluding the first two polysome fractions and one flow cytometry population with low library yields and a high degree of bottlenecking ("Methods"), 96% of variants (3396/3527) were detected in all biological replicates and readouts, enabling analysis of multi-layer effects of gene expression. 60.6% of these variants are within ROI 1 (2059), consistent with the targeted mutagenesis of more codons in ROI 1. While variant frequencies were highly reproducible between replicates of DNA and RNA readouts (>0.9 Pearson correlation), we observed only modest reproducibility for polysome and flow cytometry readouts (~0.5–0.8) (Fig. EV4C, "Discussion"). For example, in ROI 1 and 2 respectively, the mean Pearson correlation of variant frequencies was 0.61 and 0.72 for polysome fraction 3 and 0.51 and 0.80 for fraction 4. Poorer reproducibility for these readouts is possibly due to more limited starting material. After summarizing biological replicates ($N = 4$) for each readout, correlations between gDNA, total RNA, and polysomal RNA readouts were satisfactory (Pearson's $r > 0.7$). Flow cytometry readouts were the least correlated with other readouts as expected due to the different modality of measurement (Fig. EV4D).

## *COMT* variant effects on RNA abundance are highly position- and domain-specific

Given that variant frequencies are measured using sequencing, we used a compositional data analysis framework (ALDEx2) (Fernandes et al, 2013, 2014; Gloor et al, 2016; Gloor, 2021). All reported effect sizes from ALDEx2 correspond to the median standardized difference between groups ("Methods"). While ALDEx2 effect sizes are not equal to log fold changes as estimated using limma analysis ("Methods"), they are highly correlated (0.93 Pearson correlation, Fig. EV5A).

We first identified *COMT* coding variants with differential abundance at the RNA level relative to gDNA (Fig. 2B, "Methods," FDR < 0.1). Of variants with altered RNA abundance, we detected predominantly nonsense and missense variants, including a single silent variant (V63V). Specifically, 87% of nonsense variants (47/54) were significantly down in total RNA relative to gDNA, and our point estimate for nonsense variant effects on *COMT* RNA abundance was approximately a two-fold decrease relative to the gDNA frequency (fold change of 0.43, standard deviation 0.13;

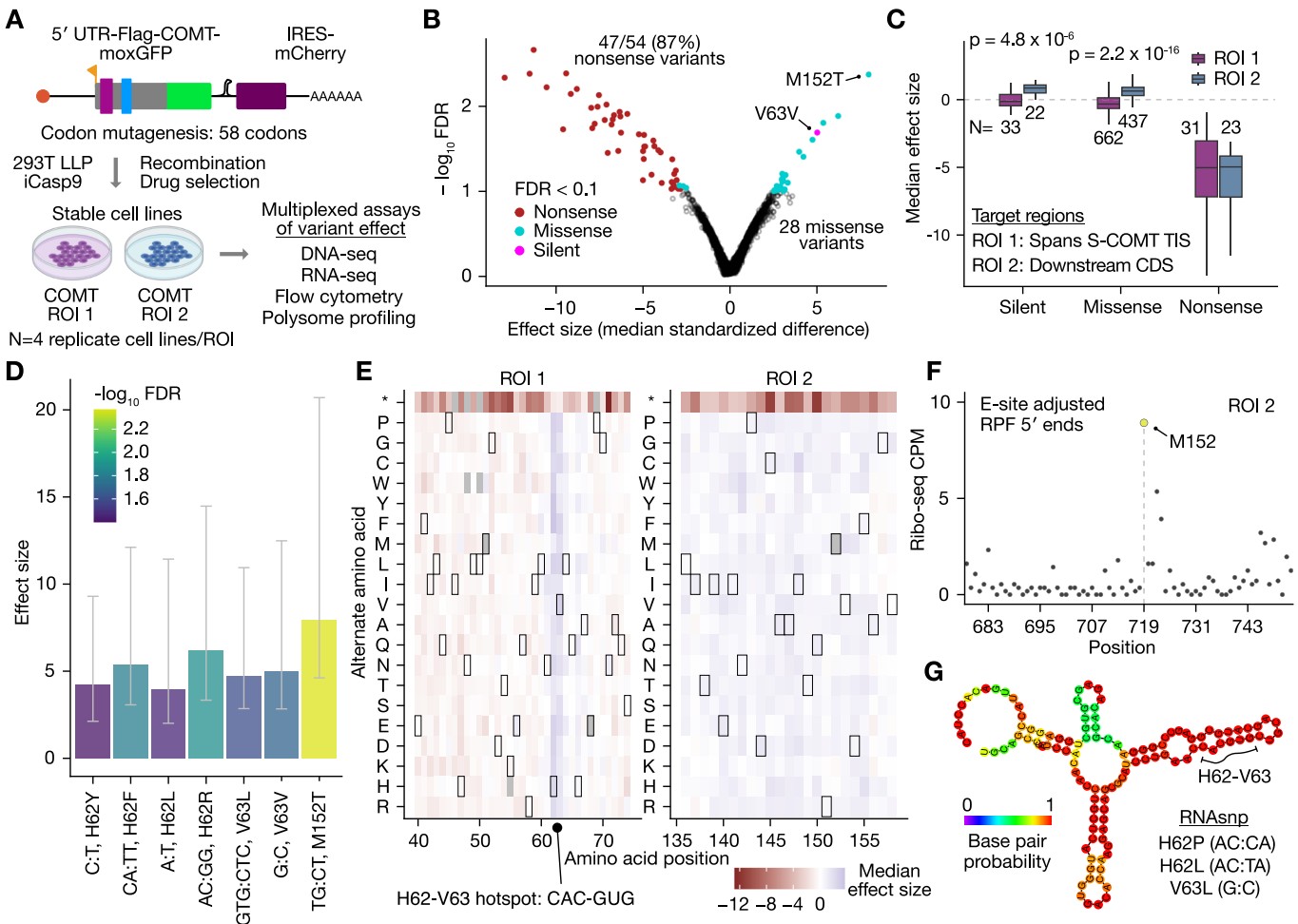

**Figure 2. Design of integrated multiplexed assays and COMT variant effects on RNA abundance.**

In (B–E), the effect size is determined by ALDEx2 (Fernandes et al, 2013, 2014; Gloor et al, 2016; Gloor, 2021) from four biological replicates ("Methods"). (A) COMT mutagenesis of two regions of interest (ROIs). Two regions spanning common population and clinically relevant SNPs were mutagenized to generate 61/63 of alternate codons at each of 58 codon positions. (B) COMT variant effects on RNA abundance. (C) RNA abundance effects differ by COMT ROI. The median effect size of synonymous variants was taken prior to plotting. The boxplot range is from the first to third quartile, center is the median, whiskers extend to the largest and smallest values no greater than 1.5 * interquartile range, and outliers beyond the whiskers were omitted for clarity. Significance between ROIs was determined by two-sided Wilcoxon tests. (D) Effects by variant position. The reference and alternate nucleotides and amino acid position are denoted for variants at a FDR < 0.05. Bars represent ALDEx2 effect size estimates by variant position and error bars indicate ALDEx2 75% confidence intervals ($N = 4$ biological replicates). (E) Position-specific effects on RNA abundance. Gray tiles indicate incomplete observations. Tiles with a border are the wild-type amino acid. (F) Potential ribosome pause with M152 in the E-site. Plotted are counts per million deduplicated reads (CPM) of E-site adjusted ribosome-protected fragments (RPFs, "Methods"). (G) Predicted local secondary structure at H62-V63. Minimum free energy structure ("Methods") of a 120 nt sequence spanning the H62-V63 hotspot. Bottom right: variants significant at a FDR < 0.1 with a predicted effect on secondary structure (RNAsnp p value < 0.1, "Methods") are shown.

"Methods"). These observations suggest that our sequencing-based estimate of variant effects on mRNA abundance has both sufficient power to detect changes and low systematic bias.

Interestingly, although in ROI 1 the majority of missense variants decreased RNA abundance, in ROI 2, most silent and missense variants increased in RNA abundance (Fig. 2C). Yet, nonsense variant effects were highly similar between the two ROIs, indicating a potential biological origin of the observation. Other than M152T, which exhibited the largest increase in RNA abundance, variants with significant effects on RNA abundance were overwhelmingly located in two adjacent codons: H62 and V63. At these codons, both silent and missense variants uniformly led to increased RNA abundance (Fig. 2D). Moreover, when

aggregated at the level of amino acid, increased RNA abundance was observed for nearly all alternate amino acids (Fig. 2E). Notably, the rs4633 (C > T) silent variant is in the wobble position of H62, and this codon is located in the stable stem structure of COMT exon 3 (Fig. 1A). In HEK293 cells, rs4633 increased protein abundance for the low pain sensitivity haplotype (Tsao et al, 2011). While Tsao et al suggested that the rs4633 T allele increases protein abundance by relaxing secondary structure near the TIS to facilitate translation initiation, our data supports a potential mechanism of increased RNA abundance (Appendix Note).

We then turned to investigating potential mechanisms causing altered RNA abundance, including differences in codon stability coefficients (Narula et al, 2019; Wu et al, 2019), ribosome dwell

time (Narula et al, 2019), dicodons (Gamble et al, 2016), dipeptides (Burke et al, 2022), tRNA abundance (Gogakos et al, 2017; Behrens et al, 2021), RNA binding protein motifs (Benoit Bouvrette et al, 2020), transcription factor binding sites (Tan and Lenhard, 2016), RNA editing sites (Picardi et al, 2017), and predicted variant effects on folding (Sabarinathan et al, 2013a). Our analysis indicated no significant global differences between variants that increased and decreased in RNA with respect to most of these potential mechanisms (codon stability coefficients, dwell time, dicodons, dipeptides, tRNA abundance, RBP motifs, RNA editing; "Methods").

While there were no global differences, specific variants highlight potential mechanisms altering mRNA abundance. For example, we examined our ribosome profiling data and noted a potential ribosome stall when M152 is situated in the E-site (Fig. 2F). We hypothesize the M152T variant (ATG:ACT) might perturb this putative stall and stabilize the RNA. This effect was specific to the ACT codon, as all other variants to threonine at this position showed no effect. Unlike M152, we did not find any strong footprint signals at the H62 or V63 codons. While the H62 codon (CAC) has a low codon stability coefficient, V63 (GUG) has an optimal coefficient (Wu et al, 2019; Narula et al, 2019), suggesting the effect at this hotspot is not solely due to changes in codon stability coefficients. We noted the H62-V63 hotspot is positioned in a high-probability stem structure (Figs. 1A and 2G). Using RNAsnp (Sabarinathan et al, 2013a), we found three variants at the hotspot that were predicted to alter RNA structure (RNAsnp $p < 0.1$, "Methods"). Taken together with prior computational evidence for a role of RNA structure (Nackley et al, 2006; Tsao et al, 2011), we hypothesize variants at H62 and V63 alter RNA folding, which may have a secondary impact on RNA abundance. However, we also note that the H62-V63 codons constitute a E-box transcription factor motif, and variants at this location were predicted to abrogate a HIF1-α/β binding site (Dataset EV1, "Methods"), which may be an alternative albeit unlikely mechanism given the distance of these sites to the promoter region.

## Flow cytometry nominates *COMT* variants with specific effects on protein abundance

Next, we determined which *COMT* variants alter protein abundance by flow cytometry. We attempted to separate RNA and protein effects by identifying variants that specifically alter moxGFP fluorescence (protein abundance) given a fluorescent proxy for RNA abundance (mCherry). Cells with a moxGFP-mCherry relationship off the diagonal line are interpreted to harbor variants that lead to protein abundance that differs from what would be expected given the inferred RNA abundance. We sorted cells from each library into four populations with differing moxGFP and mCherry fluorescence ("Methods"). Representative data for one biological replicate of the wild-type control and *COMT* variant libraries is shown in Fig. 3A. We observed high correlation between moxGFP (protein) and mCherry (RNA proxy) fluorescence (Pearson's $r$ 0.97, 0.91, and 0.85 for the wild-type template, ROI 1, and ROI 2, respectively). We note that our design is unable to unequivocally determine if protein abundance effects reflect underlying effects on RNA abundance.

To test the reliability of our flow cytometry data, we first determined the relative abundance of silent and nonsense variants in specific populations of cells (Fig. 3A). We defined variants with low

protein abundance as those enriched in population 3 (P3) ("Methods"). We found nonsense variants were enriched in the low-protein population (P3) but depleted in the high-protein population (P1) (Fig. 3B). Conversely, silent variants were depleted from P3 but were well-represented in P1. Eleven variants were enriched in P3 at a FDR < 0.05 (Fig. 3C). Aside from H62, codons where variants lead to low protein abundance overwhelmingly encoded a nonpolar, uncharged amino acid (G, A, V, L, I), suggesting hydrophobicity in the early coding region (S-COMT first and fifth helix, second β-strand) may be important for COMT folding (Fig. 3D,E). We examined variants at these positions to determine if specific chemical properties of the alternate amino acid may correlate with the direction and magnitude of effect. While changes from nonpolar amino acids to charged or polar amino acids were the predominant variants leading to low protein abundance, some of these variants were changes from a smaller nonpolar amino acid to a bulkier nonpolar amino acid (A156I, G157I) (Fig. 3F).

We then investigated the relationship of RNA and protein abundance. We found that variants down at the RNA level were depleted from P1 and enriched in P3, suggesting that some variant effects on protein abundance stem from underlying changes to RNA abundance (Fig. 3G). Similarly, variants with low RNA abundance were enriched in P2 relative to P1 (Fig. EV5B). We finally asked whether variants exhibiting low protein abundance overlapped evolutionarily constrained residues. We used DeMask (Munro and Singh, 2020) and ESM1b (Brandes et al, 2023), models for variant effect prediction, to test for a difference between variants enriched or depleted in P3. Amino acid substitutions with low DeMask scores are at conserved residues and/or show lower fitness in other multiplexed assay datasets, and the ESM1b score reports the probability of observing a particular alternate amino acid given its sequence context. We found variants enriched in P3 had lower DeMask scores compared to variants depleted in P3, reinforcing the structural importance of conserved residues in COMT (Fig. 3H). Consistently, variants enriched in P3 had lower ESM1b scores than variants depleted in P3 (Fig. EV5C).

Altogether, we mapped the effect of 3070 variants on COMT protein abundance and identified twelve constrained amino acids. Our flow cytometry strategy may be more sensitive to variants that specifically alter protein abundance, independent of effects on RNA abundance. The positions with negative effects on protein abundance largely encoded nonpolar amino acids and made contacts with adjacent secondary protein motifs, whereas the loop nearest the *S*-adenosyl methionine binding site was highly tolerant to mutation except for a single amino acid substitution (N142P). While we assume the majority of these variants decrease protein abundance by causing protein misfolding and proteasomal degradation (Larsen et al, 2023), or decreasing RNA abundance, another possibility is that they alter protein abundance by impacting translation efficiency.

## A MAVE using polysome profiling identifies *COMT* variant effects on ribosome load

We reasoned that some effects on RNA or protein abundance could be attributed to impacts on translation, via alteration of elongation or co-translational protein folding (Kimchi-Sarfaty et al, 2007; Simhadri et al, 2017). To our knowledge, no prior study measured ribosome load for codon mutagenesis libraries in native coding

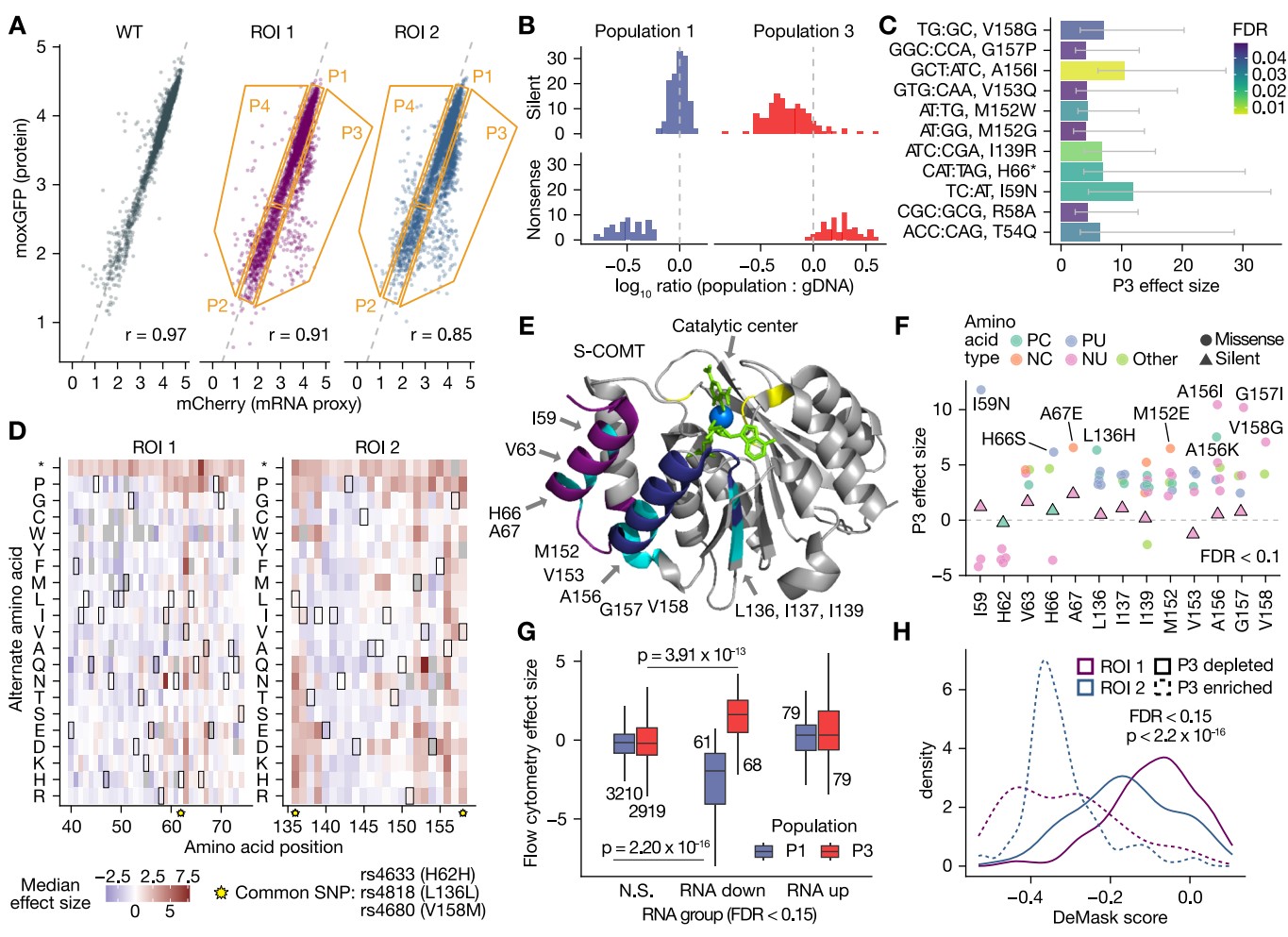

**Figure 3. COMT variant effects on protein abundance.**

In (C–G), the effect size is determined by ALDEx2 from four biological replicates ("Methods"). (A) Flow cytometry gating method. A representative biological replicate is shown. Orange lines represent the gating strategy. (B) Silent and nonsense variants are enriched and depleted in expected populations. (C) Top variants leading to low protein abundance. Variants enriched in P3 compared to gDNA are shown. Error bars indicate ALDEx2 75% confidence intervals ($N = 4$ biological replicates). (D) COMT protein variant effect map. Gray tiles indicate incomplete observations. Tiles with a border are the wild-type amino acid. Yellow stars indicate locations of common population SNPs. (E) Positions with low protein abundance variants in the S-COMT structure, with annotations relative to MB-COMT. ROIs 1 and 2 are indicated by dark purple and dark blue, respectively. Constrained positions are highlighted in cyan. Substrates (3-,5-dinitrocatechol, S-adenosylmethionine) are shown in lime green, the catalytic $Mg^{2+}$ ion in blue, and catalytic residues in yellow. Note mutational tolerance in the loop near the catalytic center. (F) Missense variant effects at constrained positions. The median effect size is plotted for each amino acid substitution. Variants at positions in (E) are colored according to the property of the alternate amino acid. PC positive charged (R, H, K); NC negative charged (D, E); PU polar uncharged (S, T, N, Q); Other (C, P); NU nonpolar uncharged (G, A, V, L, I, M, F, Y, W). (G) Comparison with RNA abundance. Variants in P1 and P3 are stratified by their effect on RNA abundance at FDR < 0.15. The boxplot range is from the first to third quartile, center is the median, whiskers extend to the largest and smallest values no greater than 1.5 * interquartile range, and outliers beyond the whiskers were omitted for clarity. Numbers next to each boxplot indicate the number of variants. (H) Comparison with linear model of protein abundance effect. Kernel density estimate of the DeMask model score is shown for variants in P3 at a FDR < 0.15. In (G, H), significance was determined with two-sided Wilcoxon tests. Source data are available online for this figure.

sequences, although previous studies have profiled ribosome load on transcript isoforms (Floor and Doudna, 2016; May et al, 2023) or randomized 5′ UTRs (Sample et al, 2019).

We sequenced RNA from several polysome *metafractions* (pooled raw fractions) for the *COMT* variant libraries and compared abundance in each metafraction to the total RNA abundance (Fig. EV5D, "Methods"). Note that since our readouts of ribosome load reflect the abundance of a variant in a particular polysome fraction, we expect a correlation between the abundance of a variant in total RNA and in polysome fractions. By comparing

the frequency between these two readouts, we can determine which variants are enriched or depleted in a given polysome fraction.

Replicates were modestly correlated for variant frequencies between heavy-polysomal metafractions (Pearson correlation mean across replicates: 0.61 and 0.72 in metafraction 3, and 0.51 and 0.80 in metafraction 4 for ROIs 1 and 2, respectively; corresponding standard deviations: 0.09, 0.23, 0.11, 0.16). Poor correlation was observed for the first and second metafractions, likely because these libraries were very low yield (Fig. EV5E). Consistent with previous measurements in HEK293T cells (Floor and Doudna, 2016), *COMT*

associates predominantly with heavy polysomes, leading to low abundance of messages with 1–3 loaded ribosomes (Fig. EV5F). High polysome load may explain low yield for the first two metafractions. Thus, we dropped these metafractions from subsequent analyses.

We hypothesized nonsense variants should be depleted in mid (F3) to heavy (F4) polysome metafractions relative to silent variants for at least two reasons: (1) shortening of the open reading frame would limit the accommodation of ribosomes, and (2) a less likely possibility is that nonsense-mediated decay (NMD) may lead to dissociation of ribosomes from the message, though lack of exon-junction complex deposition on mRNAs due to an absence of introns in our transgene should limit the efficiency of NMD (Lloyd

et al, 2020). Indeed, we found nonsense variants showed lower enrichment in F3 and F4 compared to silent variants (Fig. 4A, two-sided Wilcoxon rank sum test $p = 0.07$ and $p = 5.26 \times 10^{-9}$).

We proceeded by identifying variants with altered mid- (F3) and heavy- (F4) polysome loading. We identified 21 variants with differential abundance in F3 and F4 (Fig. 4B, FDR < 0.05). We uncovered variants at positions previously identified as significant in our RNA and protein abundance assays, including V63V (increased RNA, increased ribosome load). We cross-referenced our ribosome profiling data and noted potential ribosome pauses at M51, N71, Q73, and S74 of MB-COMT, the location of 24% (5/21) of variants with altered ribosome load (Fig. 4C). Ribosome pauses were observed for these variants when positioned in either the A-,

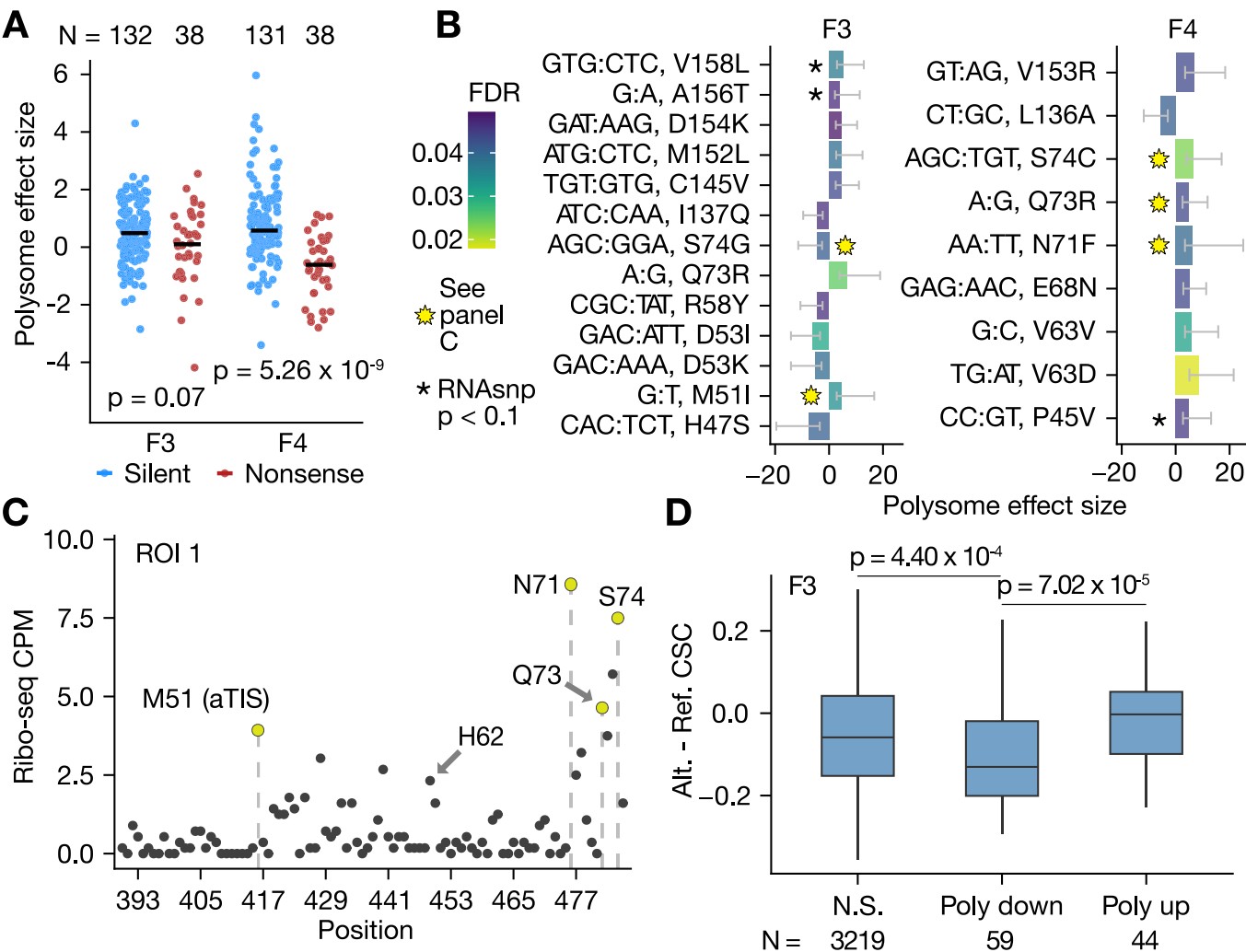

**Figure 4. *COMT* variant effects on ribosome load.**

In (A, B), the effect size is determined by ALDEx2 from four biological replicates ("Methods"). (A) Nonsense variants have decreased ribosome load relative to silent variants. F3 and F4 refer to mid to heavy polysome metafractions, respectively. Significance was determined by two-sided Wilcoxon rank sum tests. See polysome profile and fraction pooling strategy in Fig. EV5D. (B) Variants with differential ribosome load at a FDR < 0.05. Variants at putative pause sites are demarcated with a yellow star. Error bars indicate ALDEx2 75% confidence intervals ($N = 4$ biological replicates). (C) Putative pause sites in *COMT*. The y-axis reports PCR-deduplicated counts per million (CPM) of ribosome-protected fragments (RPFs) in the *COMT* template. RPF 5′ ends were codon adjusted ("Methods"), and vertical dashed lines highlight positions of differential variants. M51 is the alternative TIS (aTIS) for S-COMT. (D) Variant effects on ribosome load are consistent with codon stability coefficients (CSCs). The difference between alternate and reference CSCs ("Methods") (Narula et al, 2019) was computed for F3 variants at a FDR < 0.1. Significance was determined by a one-sided Wilcoxon test. The boxplot range is from the first to third quartile, center is the median, and whiskers extend to the largest and smallest values no greater than 1.5 * interquartile range. Source data are available online for this figure.

P-, or E-sites, indicating both codon identity and amino acid identity are important for ribosome pausing (Ingolia et al, 2011; Zhang et al, 2017; Narula et al, 2019). The majority of these variants (4 of 5) showed increased ribosome load, but our observational result is not sufficiently powered to establish a statistical relationship between ribosome pausing and load.

We finally asked whether codon stability coefficients (CSCs) or prior ribosome pausing data (Narula et al, 2019; Wu et al, 2019) provide a mechanistic interpretation for variants with altered ribosome load. Variants with low CSCs were depleted from mid-polysomes compared to variants with high CSCs (Fig. 4D, F3, one-tailed Wilcoxon rank sum test $p = 7.02 \times 10^{-5}$ and $8.73 \times 10^{-5}$ for independent datasets reporting CSCs (Narula et al, 2019; Wu et al, 2019)). Thus, in addition to CSCs as a predictor of mRNA stability and translational efficiency (protein/RNA abundance) (Presnyak et al, 2015; Bazzini et al, 2016; Mauger et al, 2019), variants that alter CSCs may also alter the number of ribosomes loaded on a mRNA. We did not find a global association between polysome effect size and tRNA abundance, or transcriptome-wide scores for ribosome dwelling at the A-site or P-site (Narula et al, 2019).

In summary, we tested the application of polysome profiling for multiplexed assay readout for thousands of *COMT* variants. We find less than 1% of queried variants (21/3341) altered mid- to heavy-polysome load (FDR < 0.05), and few of these variants were detected at the RNA- and protein-level. Our results highlight the utility of polysome profiling readouts to complement RNA and protein abundance measurements for multiplexed assays of variant effect.

## Integrated analysis of *COMT* post-transcriptional gene expression

Our analyses identified variants with effects on specific molecular phenotypes, generating a rich dataset enabling us to investigate feedforward relationships between RNA abundance, ribosome load, and protein abundance. We first cross-referenced expression quantitative trait loci (eQTL) data (Kwong et al, 2022) and population variant data from gnomAD (Karczewski et al, 2020) to determine if any common variants might impact *COMT* gene expression. While none of the variants we targeted were previously identified as eQTLs, our analyses nominate specific variants (FDR < 0.15) observed in human populations that might impact expression (Dataset EV2).

For example, we identified rs4633 (H62H, C > T), rs201641100 (P143P, C > T), rs8192488 (A146A, C > T), and a variant H62Y (C > T) that increased RNA abundance, as well as L49V (C > G) which decreased RNA abundance. At the level of ribosome load, we identified rs758271838 (L60L, C > T), rs149909767 (G70R, G > A), and rs759340648 (D144N, G > A) that increased ribosome load, and rs750684179 (G52G, T > C) which decreased ribosome load and was predicted to alter RNA secondary structure. Finally, of variants that may lead to low protein abundance, we identified rs1342715506 (I59I, C > T), rs757163626 (A67V, C > T), rs375558228 (A67A, G > A), and rs567862500 (A147T, G > A).

We next proceeded by utilizing machine learning methods to map the similarity of gene expression signatures for thousands of variants across layers. We conducted an analysis using our single-layer effect sizes (ALDEx2; Fernandes et al, 2013, 2014; Gloor et al, 2016; Gloor, 2021; "Methods") to map sequential expression

conditioned on the prior layer (RNA abundance given gDNA abundance, ribosome load given RNA, and protein given RNA). Nonsense variants primarily had an effect on RNA abundance, with modest additional effects on protein abundance and ribosome load (Fig. 5A, "Methods"). We then investigated variants that were differential in any gene expression layer and hierarchically clustered these variants (Fig. 5B, "Methods"). Notably, several variants led to divergent expression signatures (effect in different directions between layers) including at V63, M152, and L136.

Then, we used compositional data analysis (interquartile-range log ratios) (Greenacre, 2021) and an unsupervised machine learning method (self organizing map) (Kohonen, 1982; Wehrens and Buydens, 2007) to group *COMT* variants by their expression signature ("Methods"; Fig. 5C,D). Of note, three clusters (1, 2, and 9) contained variants with low protein abundance; cluster 9 contained variants with low RNA and protein abundance; and variants in cluster 8 exhibited decreased ribosome load with little to no changes on RNA or protein abundance.

We characterized clusters in terms of their composition of silent, missense, and nonsense variants. Strikingly, cluster 9 was highly enriched for nonsense variants and contained 19 missense variants and no silent variants (Fig. 5E). Clusters further away from this cluster showed few nonsense variants but an increase in the proportion of silent variants. A large proportion (14/19) of the missense variants in cluster 9 were at position I59 and A67. Although I59 is thought to be relatively tolerant to mutation, A67 is constrained (DeMask (Munro and Singh, 2020) entropy <20% percentile). These findings implicate that these specific missense variants lead to gene expression profiles highly similar to nonsense changes.

We finally tested whether the variants which show signatures similar to nonsense variants might alter RNA folding, as previously suggested (Nackley et al, 2006; Mauger et al, 2019). A predicted alteration of RNA folding was tested for 19 missense variants from cluster 9 and 11 silent variants from adjacent clusters 7 and 8 (Fig. 5F). Of these 30 variants, four (13%) were predicted to alter RNA folding (RNAsnp $p < 0.1$). For example, the silent variant E56E (G > A) was among this set, but V63V (G > C) was not predicted to significantly alter folding. We also found that variants with low protein abundance were associated with greater predicted RNA folding effects (Fig. 5G, two-sided Wilcoxon rank sum test $p = 5.61 \times 10^{-10}$). Thus, perturbed RNA folding may alter translation initiation or elongation rates with consequent impacts on protein abundance (Mauger et al, 2019).

## Discussion

Our results for variant effects on multiple gene expression phenotypes are collated and provided for each multiplexed assay, along with annotations of codon stability coefficients, tRNA abundance, ribosome dwell time, and predicted effects on RNA structure (Datasets EV3 and EV4). To our knowledge, we are the first to integrate three multiplexed assays of variant effect to measure RNA abundance, ribosome load, and protein abundance at single-nucleotide resolution in coding sequences. We coalesced multiplexed data from several gene expression layers and identified elements controlling expression in both a directional and divergent manner. Variants impacting multiple layers were rare,

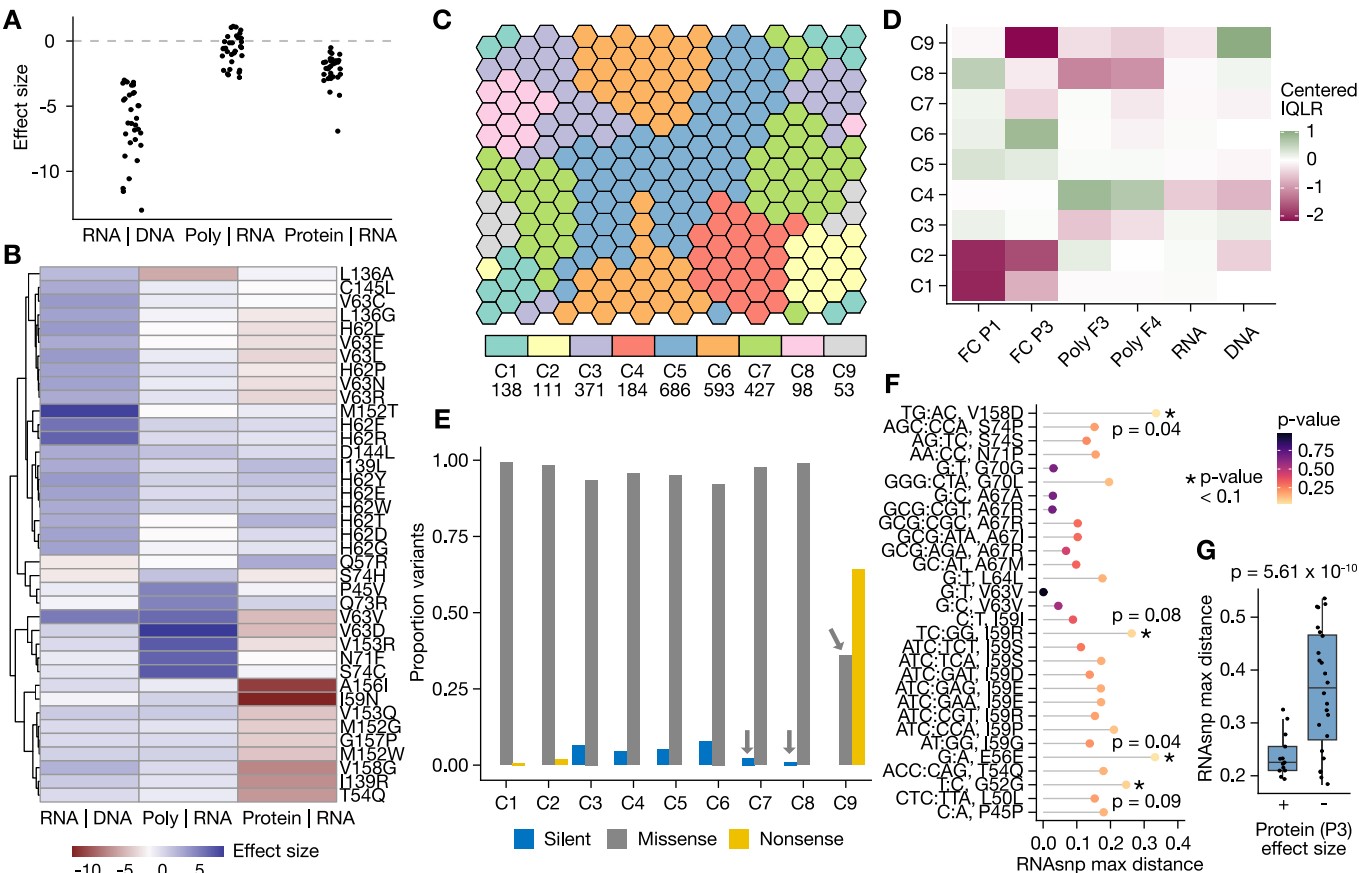

**Figure 5. Integrated analysis of *COMT* multi-layer gene expression.**

In (**A, B, G**), the effect size is determined by ALDEx2 from four biological replicates ("Methods"). In all panels, the "protein | RNA" effect sizes (enrichment in flow cytometry P3) were multiplied by −1 to signify low protein abundance, as was FC P3 in panel D. For "Poly | RNA," metafraction 4 was used. (**A**) Nonsense variants predominantly impact RNA abundance. Effect sizes are shown for 32 nonsense variants tested in all expression layers. (**B**) Hierarchical clustering of variants. Nucleotide changes were omitted for clarity. (**C**) Clustering of variant expression topology. A self-organizing map (SOM) was trained and each of 260 neurons (hexagons) were subsequently grouped by affinity propagation clustering ("Methods"). Nearby SOM neurons share a similar expression topology. (**D**) Exemplars for each affinity propagation cluster. The interquartile log ratios (IQLR) are shown, indicating the representative expression. (**E**) Variant composition of affinity propagation clusters. The proportion of variants that are silent, missense, and nonsense in each cluster are plotted. Arrows indicate variants tested in (**F**). (**F**) Predicted effects on RNA structure for missense variants in cluster 9 and silent variants in clusters 7 and 8. Variants with a potential impact on RNA structure were determined with RNAsnp ("Methods"). Variants impacting structure have greater distances. (**G**) Alteration of RNA structure may underlie protein abundance effects. Variants affecting protein abundance at a FDR < 0.1 and RNAsnp *p* value < 0.15 are plotted. Significance was determined with a two-sided Wilcoxon rank sum test. The boxplot range is from the first to third quartile, center is the median, and whiskers extend to the largest and smallest values no greater than 1.5 * interquartile range.

underscoring the benefit of harnessing several multiplexed assays to annotate variant phenotypes. Our data suggest that variant effects may be explained by diverse mechanisms including changes in codon stability coefficients and perturbation of RNA structure or ribosome pausing.

Prior studies reported a strong correlation between ribosome load and protein abundance (0.57-0.87 Pearson correlation) (Floor and Doudna, 2016; Sample et al, 2019), which prompted us to utilize polysome profiling as the basis of a novel MAVE to measure ribosome load. Silent variants in *CFTR* and *F9* may decrease protein abundance through alterations to translation elongation, leading to cystic fibrosis and Hemophilia B, respectively (Kirchner et al, 2017; Simhadri et al, 2017). In addition, SNPs increased the proportion of disome versus monosome footprints in ribosome profiling data, suggesting some variants may alter elongation by causing ribosome stalling (Li and Chen, 2023). Our variant library

enabled us to test the effect of silent variants on *COMT* translation, and we found a silent variant at V63 that increased both RNA abundance and heavy polysome load.

The reliability of polysome profiling in reporting active translation is debated (Richter and Coller, 2015). Some mRNAs are polysome-associated but translationally repressed (Richter and Coller, 2015; Clark et al, 2000; Braat et al, 2004; Andrews et al, 2011). Although we identified several variants that alter ribosome load, some coinciding with putative ribosome pause sites (Fig. 4B,C), we found no strong associations between ribosome load and protein abundance for the variants tested. This may be due to our analysis strategy, which conditioned effects of ribosome load and protein abundance on underlying RNA abundance. This result is in contrast to a recent study that found a relatively high correlation between polysome load and protein levels for a library of 5′ UTRs in yeast (May et al, 2023). Potential explanations for this

discrepancy include differences in expression constructs and analysis strategies for ribosome load and protein abundance readouts (Appendix Note).

Characterization of *COMT* variants nominated mutations at hydrophobic residues that decrease protein abundance (MB-COMT residues I59, V63, A67, L136, I137, I139, V153, A156, G157, and V158). By pairing our data with predicted effects on RNA folding, we found variants at P45, G52, E56, I59, H62, V63, A156, and V158 that may alter gene expression through perturbation of RNA structure. In addition, we found that variants with low protein abundance led to a greater alteration of RNA secondary structure (Fig. 5G). Notably, a recent study indicated that increased secondary structure in the coding sequence synergizes with codon optimality to increase protein abundance (Mauger et al, 2019).

Our discovery of the H62-V63 RNA abundance hotspot is consistent with prior reports of a compensatory effect of rs4633 (H62 silent mutation) on decreased protein abundance (Tsao et al, 2011). Our results prompt future studies in *COMT* utilizing multiplexed assays of variant effect on RNA folding (Cheng et al, 2017; Kladwang et al, 2011; Kladwang and Das, 2010) using mutational profiling RNA probing methods (Zubradt et al, 2017; Weng et al, 2020).

To fully realize the power of integrative multiplexed assays, readouts of protein function are needed in combination with gene expression measurements. We attempted to establish a high-throughput assay of COMT enzyme activity using a COMT substrate (7,8-dihydroxy-4-methylcoumarin, DHMC), which was suggested to emit fluorescent signal upon methyl transfer (Qian et al, 2016). However, we failed to detect activity of endogenous COMT after treating HEK293T landing pad cells with DHMC, including after expression of our transgene in multiple experiments. Our data suggests the reporter may be nonfunctional, highlighting the need to develop more rigorously tested reporters of enzyme activity.

There are several important caveats in our study that need to be considered when interpreting the findings. First, our *COMT* transgene is expressed without introns, with partial vector-derived UTRs, and with N- and C-terminal affinity tags, which could impact its expression relative to the endogenous *COMT* gene (Appendix Note). Second, we included an IRES-mCherry element to enable flow cytometry analysis, which would impact the observed ribosome load on our construct. This may limit the sensitivity to detect variants causing subtle changes to translation.

Third, high ribosome load on our transgene, combined with our polysome fraction pooling strategy (in which the first two metafractions did not produce sufficient yield), led to a limited dynamic range. This may lead to lower reproducibility of polysome profiling readouts compared to readouts from gDNA and total RNA, which could be a limitation for scaling multiplexed translation assays to large targets. Potential reasons for modest reproducibility of variant abundance (Fig. EV4) include biological variability in translation and a technically challenging experimental workflow (three-step RNA purification with >30 RNA precipitations *per metafraction*, "Methods"). Robotic automation of RNA purification steps and use of scalable library preparation methods (Zheng et al, 2014) might be leveraged for multiplexed polysome profiling of large target genes. For future high-throughput polysome profiling assays, use of stable-expression tools, a carefully chosen polysome fraction pooling design, and a relatively high number of biological replicates (Matreyek et al, 2017, 2020) may be required.

Taken together, these limitations may explain our finding of insignificant effects of common population SNPs (Appendix Note). This includes the rs4818 variant, which generates the high-pain sensitivity haplotype and was previously reported to lead to low protein abundance (Nackley et al, 2006). Nonetheless, we identified the encompassing residues of rs4818 (I136) and rs4680 (V158) as positions intolerant to mutation by our multiplexed protein abundance assay.

Our study underscores the advantages of employing comprehensive gene expression profiling for rare variants, a strategy that revealed several variants in human populations potentially influencing *COMT* gene expression. Further research will be required to explore the functional implications of rare variants on expression phenotypes for *COMT* and other clinically relevant genes.

# Methods

## Reagents and tools

See Table 1.

**Table 1. Reagents and tools.**

| Reagent/resource | Reference or source | Identifier or catalog number |
|---|---|---|
| Experimental models | | |
| HEK293T LLP iCasp9 Blast | (Matreyek et al, 2020) | Fowler Lab, The University of Washington |
| Endura Electrocompetent cells | Lucigen | 60242 |
| Recombinant DNA | | |
| COMT in pLDNT7_nFLAG | DNASU | HsCD00617865 |
| pDEST_HC_Rec_Bxb_v2 | This study | Roth Lab, The University of Toronto |
| pCAG-NLS-HA-Bxb1 | This study | Roth Lab, The University of Toronto |

**Table 1.** (continued)

| Reagent/resource | Reference or source | Identifier or catalog number |
|---|---|---|
| pDONR223_UTR5_Flag_COMT_moxGFP_LPS | This study | N/A |
| pDEST_HC_Rec_Bxb_v2_UTR5_Flag_COMT_moxGFP_LPS | This study | N/A |
| pDEST_HC_Rec_Bxb_v2_UTR5_Flag_COMT_moxGFP_LPS_ROI1 | This study | N/A |
| pDEST_HC_Rec_Bxb_v2_UTR5_Flag_COMT_moxGFP_LPS_ROI2 | This study | N/A |
| pDEST_HC_Rec_Bxb_v2_UTR5_Flag_COMT_moxGFP_LPS_uORF_A_mutant | This study | N/A |
| pDEST_HC_Rec_Bxb_v2_UTR5_Flag_COMT_moxGFP_LPS_uORF_B_mutant | This study | N/A |
| pDEST_HC_Rec_Bxb_v2_UTR5_Flag_COMT_moxGFP_LPS_uORF_B_restore_mutant | This study | N/A |
| pDEST_HC_Rec_Bxb_v2_UTR5_Flag_COMT_moxGFP_LPS_Flag_deletion_mutant | This study | N/A |
| pDEST_HC_Rec_Bxb_v2_UTR5_Flag_COMT_moxGFP_LPS_PCBP_deletion_mutant | This study | N/A |
| pDEST_HC_Rec_Bxb_v2_UTR5_Flag_COMT_moxGFP_LPS_UTR5_deletion_mutant | This study | N/A |
| Antibodies | | |
| Mouse monoclonal anti-Flag M2 | Sigma Aldrich | F3165 |
| Mouse monoclonal [AC-15] to beta-Actin | Abcam | Ab6276 |
| Goat anti-mouse IgG (H + L) Highly Cross-Adsorbed Secondary Antibody, Alexa Fluor Plus 647 | Thermo Fisher Scientific | A32728 |
| Oligonucleotides and other sequence-based reagents | | |
| PCR primers | This study | Dataset EV5 |
| NEBNext Multiplex Oligos for Illumina Dual Index Primers Set 1 | New England Biolabs | E7600S |
| Chemicals, enzymes, and other reagents | | |
| Dulbecco's Modified Eagle Medium | Thermo Fisher Scientific | 11995065 |
| Fetal bovine serum | Thermo Fisher Scientific | 10-437-028 |
| Penicillin–Streptomycin | Thermo Fisher Scientific | 15140122 |
| Q5 polymerase | New England Biolabs | M0491 |
| Nucleospin PCR and Gel Cleanup Kit | Takara | 740609 |
| Gibson Assembly Master Mix | New England Biolabs | E2611 |
| Gateway BP Clonase II Enzyme Mix | Thermo Fisher Scientific | 11-789-020 |
| Gateway LR Clonase II Enzyme Mix | Thermo Fisher Scientific | 11-791-020 |
| Zymoclean Gel DNA Recovery Kit | Zymo Research | D4007 |
| ZymoPURE II Plasmid Maxiprep Kit | Zymo Research | D4203 |
| BlpI | New England Biolabs | R0585 |
| Q5 Site-Directed Mutagenesis Kit | New England Biolabs | E0554S |
| Lipofectamine 3000 | Thermo Fisher Scientific | L3000008 |
| Doxycycline | Fisher Scientific | 50-165-6822 |
| AP1903 | MedChemExpress | HY-16046 |
| DNaseI | New England Biolabs | M0303 |
| RNaseI | Thermo Fisher Scientific | AM2294 |
| RNA Clean and Concentrator Kit | Zymo Research | R1015 |
| Random Hexamers | Fisher Scientific | N8080127 |
| SuperScript IV Reverse Transcriptase | Thermo Fisher Scientific | 18090010 |
| SUPERase-In RNase Inhibitor | Thermo Fisher Scientific | AM2696 |
| RNaseH | New England Biolabs | M0297 |
| PowerUp SYBR Green Mastermix | Applied Biosystems | A25742 |
| SYBR Gold | Fisher Scientific | S11494 |
| Pierce BCA Protein Assay Kit | Thermo Fisher Scientific | NW00080BOX |
| Pierce BCA Protein Assay Kit | Thermo Fisher Scientific | 23327 |

**Table 1.** (continued)

| Reagent/resource | Reference or source | Identifier or catalog number |
|---|---|---|
| Cycloheximide | Sigma Aldrich | C4859 |
| Ribonucleoside vanadyl complex | Fisher Scientific | 50-812-650 |
| Protease inhibitor cocktail EDTA-free | Fisher Scientific | 539196 |
| GlycoBlue | Thermo Fisher Scientific | AM9515 |
| AMPure XP beads | Beckman Coulter | A63880 |
| D-Plex Small RNA Kit | Diagenode | C0503000 |
| High Sensitivity DNA Kit | Agilent | 5067-4626 |
| Cell and Tissue DNA Isolation Kit | Norgen Biotek Corp. | 24700 |
| QIAquick PCR Purification Kit | Qiagen | 28106 |
| KAPA Library Quantification Kit | Roche | KK4873 |
| BluePippin 2% gel | Sage Science | BDF2010 |
| Software | | |
| satmut_utils v1.0.3-dev001 | (Hoskins et al, 2023; Hoskins, 2023) | |
| ALDEx2 v1.30.0 | (Fernandes et al, 2013, 2014; Gloor et al, 2016) | |
| limma v3.54.2 | (Ritchie et al, 2015) | |
| RiboFlow v0.0.1 | (Ozadam et al, 2020) | |
| RiboR v1.10.0 | (Ozadam et al, 2020) | |
| FACSDiva v6.1.3 | BD Biosciences | |
| FlowJo | BD Biosciences | |
| RNAsnp v1.2 | (Sabarinathan et al, 2013a, 2013b) | |
| TFBSTools v1.36.0 | (Tan and Lenhard, 2016) | |
| PyMol v2.5.0 | (DeLano, 2002) | |
| kohonen v3.0.12 | (Wehrens and Buydens, 2007) | |
| apcluster v1.4.11 | (Bodenhofer et al, 2011) | |
| Other | | |
| Illumina NovaSeq 6000 | Illumina | SORP |
| Illumina HiSeq X | Illumina | SORP |
| BluePippin | Sage Science | SORP |
| LSRFortessa | BD Biosciences | SORP |
| FACSAria Fusion | BD Biosciences | SORP |
| ViiA 7 Real-Time PCR System | Applied Biosystems | SORP |
| Piston Gradient Fractionator | BioComp | SORP |
| Triax Flow Cell | BioComp | SORP |
| Gradient Maker | BioComp | SORP |
| SW-41 Ti rotor | Beckman Coulter | SORP |
| UC tubes 9/16 ×3-1/2 | Fisher Scientific | NC9194790 |
| Spin X Centifuge Tube Filter with cellulose acetate membrane | Fisher Scientific | 07-200-385 |
| Falcon polystyrene round bottom tube | Fisher Scientific | 08-771-23 |
| 5PRIME Phase-Lock Gel heavy tubes | QuantaBio | 2302830 |

## Methods and protocols

### Cell culture

HEK293T LLP iCasp9 Blast cells (Matreyek et al, 2020) were periodically tested and confirmed free of Mycoplasma, and were cultured in Dulbecco's Modified Eagle Medium with 10% fetal bovine serum, and 1% penicillin–streptomycin.

### Cloning of dual-tagged COMT transgene

C-terminal fluorophore tagging of COMT has been previously reported and is compatible with enzyme activity assays (Schott et al, 2010; Sei et al, 2010). A plasmid containing the *COMT* noncoding exon 2 (NM_000754.4) and coding sequence was obtained from DNASU (clone HsCD00617865), and gBlocks were ordered for (1) the same *COMT* noncoding exon 2 with a downstream Flag tag; and (2) moxGFP (Costantini et al, 2015), which facilitates membrane localization. The 5′ UTR and coding sequence were amplified in Q5 polymerase reactions with 10 ng HsCD00617865 template and 500 nM forward and reverse primers for 25 cycles using the manufacturer's recommendations with 55 °C annealing temperature. See Dataset EV5 for primer sequences. Products were purified with the Nucleospin PCR and Gel Cleanup Kit), and the 5′ UTR, CDS, and moxGFP gBlock were joined using a Gibson assembly reaction following manufacturer recommendations.

To generate the 5′ UTR deletion clone, 2 µL of the assembly reaction was used as template for a final Q5 PCR using Gateway-compatible primers with attB1/attB2 tails and the aforementioned PCR conditions. The product was subsequently transferred into a pDONR223 entry clone via a Gateway BP reaction, and transferred to a destination vector (pDEST_HC_Rec_Bxb_v2, plasmid map available in the GitHub repository) compatible with recombination into the 293 T LLP iCasp9 Blast cell line with a 1 h LR reaction. To generate a clone containing the full 5′ UTR exon 2, first a CDS-moxGFP product was amplified from the validated 5′ UTR deletion clone in a 50 µL NEB Q5 reaction for 25 cycles with 30 s annealing at 55 °C. The product was extracted from a 1% agarose-TBE gel (Zymo Gel Extraction kit, 10 µL elution), then assembled with the 5′ UTR gBlock in a Gibson reaction. A final 50 µL Q5 PCR reaction was performed with 2 µL assembly reaction template for 30 cycles with 30 s annealing at 50 °C. The product was extracted from a 1% agarose-TBE gel as previously detailed. The transgene was transferred to the pDONR223 and pDEST_HC_Rec_Bxb_v2 with 1 h Gateway BP and LR reactions, respectively.

The isolated transgene clones contained either the full 91 nt of exon 2 of the MB-COMT 5′ UTR (Endo. 5′ UTR), or a 53 nt exon 2 deletion (5′ UTR del). Both transgenes shared a common Kozak sequence between -21-1, a N-terminal Flag tag, a C-terminal moxGFP tag upstream of a bicistronic IRES-mCherry element, and partial vector-derived 5′ and 3′ UTRs from the landing pad vector pDEST_HC_Rec_Bxb_v2. Constructs were verified using Sanger sequencing.

### COMT variant library generation

The 5′ UTR-Flag-COMT-moxGFP clone in pDEST_HC_Rec_Bxb_v2, was sent to Twist Biosciences for mutagenesis of membrane-COMT codons 40-74 (ROI 1) and 136-158 (ROI 2) using degenerate primers. All silent and missense variants and the amber nonsense variant were generated at each codon in the template with added attB recombination sites. Inserts for each target ROI were received and pooled separately, with codons in each ROI pooled at equimolar ratios. A Gateway BP reaction was performed with 70 ng of pooled inserts and 150 ng pDONR223 at 25 °C overnight (19 h). 1.5 µL BP reaction was electroporated into 25 µL Endura Electrocompetent cells with the following conditions: 2 mm cuvette, 2500 V, 200 Ohms, 25 µF. Cells were outgrown for 1 h at 37 °C in 1 mL Lucigen outgrowth media, then 600 µL was plated on Nunc Square Bioassay dishes, scraped in 8 mL LB Miller broth, and plasmid library purified with the ZymoPURE II Plasmid Maxiprep Kit using 200 µL 50 °C elution buffer, and including endotoxin removal. Library sizes were estimated at 942,000 and 510,000 clones for ROI 1 and ROI 2 libraries, respectively (or 434-fold and 357-fold coverage of each variant).

One µg of each library in pDONR223 was digested with Blp1 for 1 h 15 min at 37 °C. Digests were run on a 0.8% agarose-TAE gel and full-length plasmids were extracted with the Nucleospin PCR and Gel Cleanup Kit (Takara, 740609). A LR reaction was performed with 150 ng entry library and 150 ng pDEST_H-C_Rec_Bxb_v2, incubated at 25 °C for 21 h. 25 µL Endura Electrocompetent cells were transformed with 1.5 µL LR reaction using the same conditions as for the BP reaction. After 14.5 h, colonies were collected with 7 mL LB Miller broth, and 3 mL was pelleted and processed with the ZymoPURE II Plasmid Maxiprep Kit as above.

### Generation of single COMT mutants

The common population variants (rs4633, rs4818, rs4680, rs6267, rs74745580) were generated in the 5′ UTR-Flag-COMT-moxGFP clone in pDEST_HC_Rec_Bxb_v2 by Genscript. The 5′ UTR mutants were generated using the NEB Q5 Site-Directed Mutagenesis Kit according to the manufacturer's recommendations but extending the KLD reaction to 10 min. The cycling parameters were: initial denaturation at 98 °C for 30 s; 25 cycles of denaturation at 98 °C for 10 s, anneal at 62 °C for 30 s, extension at 72 °C for 3 min 30 s; final extension at 72 °C for 2 min. Mutagenesis primers are listed in Dataset EV5. All mutants were bidirectionally validated by Sanger sequencing.

### Stable expression of COMT variant libraries and single mutants

For each ROI 1 and ROI 2 biological replicate cell line, 20 µg of the library along with an equal mass of Bxb1 recombinase (pCAG-NLS-HA-Bxb1) was transfected into a 15 cm dish of HEK293T LLP iCasp9 Blast cells using Lipofectamine 3000, with volumes scaled based on 3.75 µL reagent per 6-well. For single mutants, 1.5 µg was transfected for each of the donor and Bxb1 recombinase plasmids. After 48 h, at near full confluency, 2 µg/mL Doxycycline and 10 nM AP1903, both solubilized in DMSO, were added for negative selection of non-recombined cells. The next day, dead cells were removed and recombined cells were grown out for an additional two days with fresh media containing Doxycycline and AP1903. Cells were recovered for two days by growth in media without Doxycycline and AP1903. Before functional assays, transcription was induced with 2 µg/mL Doxycycline for 24 h (total RNA, polysome RNA readouts) or 21 h (flow cytometry readouts), and cells were stimulated with fresh media for 1 h 15 min prior to harvest. Importantly, because our *COMT* transgene is membrane localized and might face the extracellular space, cells were collected by pipetting in cold PBS, and not by trypsinization.

### RT-qPCR

One µg of extracted total RNA and polysomal RNA was treated with DNaseI in a 100 µL reaction at 37 °C for 15 min, then re-purified by the RNA Clean and Concentrator Kit and eluted in 10 µL water. 100–200 ng of treated RNA was denatured at 65 °C for 5 min followed by primer annealing at 4 °C for 2 min, using 1 µL of 20 µM oligodT(18)6 N and 50 ng random hexamers. Primed total RNA was included in a 20 µL SuperScript IV cDNA synthesis reaction with SUPERase-In RNase inhibitor, and first-strand cDNA was synthesized by incubating at 23 °C for 10 min, 55 °C for 1 h, followed by RT inactivation at 80 °C for 10 min. RNA was digested with addition of 5 U RNaseH (NEB, M0297) to the 1st strand cDNA synthesis reaction and incubation at 37 °C for 20 min. 2 µL of a 1:10 dilution of 1st strand cDNA was assayed by qPCR on Applied Biosystems Viia-7 instruments in 10 µL reactions with PowerUp SYBR Green master mix according to the manufacturer's recommendations, with 500 nM primers and annealing at 55 °C for 15 s. See Dataset EV5 for primer sequences to the target (*COMT, mCherry*) and calibrator (*EEF2*).

### Immunoblotting

Five µg total protein, quantified by Pierce BCA Protein Assay Kit, was loaded on Bolt 8% Bis-Tris Gels in 1× LDS sample buffer with reducing conditions and addition of Bolt Antioxidant, then resolved at 200 V for 25 min. Proteins were transferred to 0.45 µm PVDF membranes at 20 V for 1 h 15 min at 4 °C. Membranes were dried for 10 min, then reactivated in methanol, and blocked for 1 h in 3% non-fat dry milk in TBST (0.1% Tween-20). Membranes were rotated with 1 µg/mL mouse anti-FLAG M2 antibody in blocking buffer overnight at 4 °C, then washed four times for 5 min in TBST. Blots were finally rotated with 0.2 µg/mL AF647-conjugated goat anti-mouse IgG secondary antibody in TBST for 1 h at RT before washing 4 × 5 min in TBST, and imaging on a Odyssey CLx instrument (700 nm detection). Densitometry analysis was conducted with ImageStudioLite.

### Polysome fractionation and polysomal RNA purification

For *COMT* library stable cell lines, cells at 80–95% confluency (one 15 cm dish per gradient) were incubated with 100 µg/mL cycloheximide in media for 10 min at 37 °C, then resuspended in ice cold PBS with 100 µg/mL cycloheximide, collected at 4 °C by pipetting, and pelleted at $300 \times g$ for 7 min at 4 °C. Pellets were flash frozen in liquid nitrogen, then lysed on ice for 10 min with 300-400 µL lysis buffer (20 mM Tris-HCl pH 7.5; 150 mM NaCl; 5 mM MgCl₂, 1 mM fresh DTT, 20 mM ribonucleoside vanadyl complex, 1× protease inhibitor cocktail EDTA-free, and 100 µg/mL cycloheximide. Lysates were clarified by centrifugation at $1300 \times g$ for 10 min at 4 °C, and 1/10th of the lysate was saved for total RNA extraction. 20–50% sucrose gradients were made with the BioComp Gradient Maker in 20 mM Tris-HCl pH 7.5, 150 mM NaCl, 5 mM MgCl₂, 1 mM DTT, and 100 µg/mL cycloheximide using Beckman Coulter UC tubes 9/16 × 3-1/2, and equilibrating overnight at 4 °C. Clarified lysates were loaded onto sucrose gradients, and fractionated by ultracentrifugation at 38,000 rpm for 2.5 h at 4 °C in a Beckman Coulter SW-41 Ti rotor.

Approximately 500 µL polysome fractions were collected using the BioComp Piston Gradient Fractionator (v8.04) with a Triax flow cell (BioComp model FC-1) and Gilson Fraction Collector using a piston speed of 0.2 mm/s. RNA was precipitated from raw polysome fractions by addition of sodium acetate to 300 mM pH 5.2, 2 µL Glycoblue, and 2x volumes cold ethanol, stored overnight at −20 °C. RNA was resuspended in 50-80 µL ultrapure water, and these raw fractions were pooled into metafractions and extracted by phenol-chloroform (5:1) and ethanol precipitation overnight at −20 °C. After a single 70% ethanol wash, polysomal RNA was resuspended in 30 µL ultrapure water. Lastly, DNaseI treatment and a column purification was performed prior to library preparation (see "RNA extraction and cDNA synthesis").

### Ribosome profiling

A 15 cm dish of cells with stable expression of the 5′ UTR-containing COMT transgene at 95% confluency was washed twice with cold PBS containing 100 µg/mL cycloheximide, then lysed on the plate with 1.5 mL lysis buffer (20 mM Tris HCl pH 7.4, 150 mM NaCl, 5 mM MgCl₂, 1 mM DTT, 100 µg/mL cycloheximide, 100 µg/mL anisomycin, 1% Triton-X, and 1× protease inhibitor cocktail EDTA-free). Lysate was partitioned into two tubes and 7 µL RNaseI was added and incubated for 1 h at 4 °C. Digestion was stopped by addition of ribonucleoside-vanadyl complex to 20 mM final. Monosomes were pelleted through a sucrose cushion (1 M sucrose, 20 mM Tris HCl pH 7.4, 150 mM NaCl, 5 mM MgCl₂, 1 mM DTT, 1x protease inhibitor cocktail EDTA-free) by spinning at 38,000 rpm at 4 °C in a SW-41 Ti rotor. Monosome pellets were resuspended in 400 µL lysis buffer and rocked overnight. RNA was extracted with 1.2 mL phenol/chloroform/isoamyl alcohol and precipitated with 300 mM sodium acetate pH 5.2, 1.5 µL Glycoblue, and 2.5 volumes of cold absolute ethanol overnight at −20 °C. RNA was pelleted at $21,000 \times g$ for 1 h, washed once with cold 70% ethanol, and resuspended with 30 µL water.

Fifteen µL of RNA was run on a 15% TBE-urea polyacrylamide gel at 150 V for 1.5 h. The gel was stained with 1× SYBR Gold and footprints between 21 nt and 34 nt were cut and isolated by crush and soak in RNA extraction buffer (300 mM sodium acetate pH 5.5, 1 mM EDTA, 0.25% v/v SDS) with overnight rotation. The eluate was passed through a Spin X Centrifuge Tube Filter with a cellulose acetate membrane, and the RNA was precipitated with 300 mM sodium acetate pH 5.2, 1.5 µL Glycoblue, and 2.5 volumes of cold absolute ethanol overnight at −20 °C. RNA was pelleted at $21,000 \times g$ for 1 h, washed once with cold 70% ethanol, and resuspended in 12 µL water. 25 ng of RNA in 8 µL water was input to the D-Plex Small RNA kit and libraries were prepared according to the manufacturer recommendations with the exception that 9 cycles of PCR were carried out. The library was purified with 1.25x AMPure beads and quantified by the High Sensitivity DNA Kit.

### Flow cytometry analysis and flow cytometry sorting

Following induction with doxycycline and media stimulation (see "Stable expression of COMT variant libraries and single mutants"), cells were washed once with cold PBS, resuspended with PBS, and assayed in a 5 mL Falcon polystyrene round bottom tube with a cell-strainer cap. Analysis of point mutants and deletion cell lines was conducted on a LSRFortessa instrument (BD) with FACSDiva v6.1.3, and flow cytometry was conducted on a FACSAria Fusion instrument (BD). Gating was performed with FlowJo. A gate for cells versus debris was defined based on FSC-H versus SSC-H, and another gate for single cells versus doublets was defined based on FSC-A versus FSC-H. Fluorescence gating was based on auto-fluorescence of 239 T LLP iCasp9 Blast cells without recombination

or of a stable cell line expressing the template transgene but without doxycycline induction. After flow cytometry sorting, each population was expanded up to a confluent T-25 flask, washed, trypsinized, pelleted, and flash frozen prior to gDNA extraction.

### COMT library preparation

gDNA extraction.  gDNA was extracted from approximately 3–4 million cells with the Cell and Tissue DNA Isolation Kit, including RNaseA treatment at 37 °C for 15 min, and eluting in 200 μL warm elution buffer.

RNA extraction and cDNA synthesis.  Approximately 3–4 million cells were solubilized with 1 mL QIAzol and 0.2 mL chloroform in 5PRIME Phase-Lock Gel heavy tubes, according to the manufacturer's recommendations. RNA was precipitated at −20 °C following the addition of 2 μL GlycoBlue and 2.5 volumes of cold absolute ethanol. RNA was washed once with cold 70% ethanol then resuspended in 30 μL water. 10 μg total RNA or 15 μL of polysomal RNA (half of 30 μL eluate, volumetric inputs) was treated with DNaseI in a 100 μL reaction at 37 °C for 15 min, then re-purified by the RNA Clean and Concentrator Kit and eluted in 20 μL water.

Ten μL DNaseI-treated total RNA (<10 μg) or DNaseI-treated polysomal RNA (200 ng–5 μg) was denatured at 65 °C for 5 min followed by RT primer annealing at 4 °C for 2 min, using 2 pmol pDEST_HC_Rec_Bxb_v2_R primer specific for the landing pad (see Dataset EV5) and 2.5 mM random hexamers. Primed total RNA was included in 20 μL SuperScript IV cDNA synthesis reaction with SUPERase-In RNase inhibitor, and first-strand cDNA was synthesized by incubating at 23 °C for 10 min, 55 °C for 1 h, followed by RT inactivation at 80 °C for 10 min. RNA was digested with addition of 5 U RNaseH to the first strand cDNA synthesis reaction and incubation at 37 °C for 20 min.

Landing pad amplification (PCR1).  Nucleic acids (gDNA, first-strand cDNA) were amplified with Q5 polymerase in 50 μL PCR reactions for 16 cycles with 500 nM landing-pad-specific primers (pDEST_HC_Rec_Bxb_v2_F/R) flanking the entire COMT insert (~1.7 kb). For negative control (plasmid) samples, 10 ng was used as template. For total RNA and polysome RNA samples, 15 μL of unpurified 1st strand cDNA synthesis reaction was used as template. For total gDNA samples, 4 μg template was used in each of three PCR reactions. For flow-sorted gDNA samples, either 1.75 μg template was used in three PCR reactions or 2.5 μg was input into two reactions, with 14 cycles instead of 16. The cycling parameters were: initial denaturation at 98 °C for 30 s; 3-step cycling with denaturation at 98 °C for 10 s, anneal at 65 °C for 30 s, extension at 72 °C for 1 min; final extension at 72 °C for 2 min.

Products from total RNA and polysome RNA samples were purified with the QIAquick PCR Purification Kit and eluted in 30 μL elution buffer. Products for total gDNA samples were pooled, resolved on a 0.8% agarose/TAE gel, stained with 1x SYBR Gold, and extracted using the Nucleospin Gel and PCR Cleanup Kit or Zymoclean Gel DNA Recovery Kit with 20 μL 70 °C elution buffer. At this point, PCR1 products for gDNA samples cannot typically be seen in the gel, and the expected band size is cut using comparison to a high-molecular weight ladder.

Coding sequence amplification (PCR2).  Purified PCR1 products (landing pad insert) were amplified for each COMT target ROI in a

50 μL NEB Q5 reaction for 8 cycles, following the same cycling parameters as for PCR1 and 500 nM tailed primers (see Dataset EV5). 15 μL PCR1 product was used as template (50% of eluate for total RNA and polysome RNA samples; 75% of eluate for gel-extracted gDNA samples). Products were purified with the Nucleospin Gel and PCR Cleanup Kit with 1:5 buffer NTI dilution and 25 μL 70 °C elution buffer, or with the QIAquick PCR Purification Kit with 30 μL elution buffer.

Illumina adapter addition (PCR3).  A final Q5 PCR reaction was carried out for 8 cycles with the same formulation as PCR2 but using NEBNext Multiplex Oligos for Illumina Dual Index Primers Set 1 according to manufacturer's recommendations (65 °C annealing). 25 μL out of 30 μL eluate was used as template for total gDNA, total RNA, and polysome RNA samples. For flow cytometry gDNA samples, 15 μL out of 25 μL eluate was used as template. PCR3 products (final library) were purified with the Nucleospin Gel and PCR Cleanup kit and eluted in 25 μL 70 °C buffer.

Quantification and size-selection of final libraries.  Final libraries were quantified using the KAPA Library Quantification Kit for Illumina according to the manufacturer's recommendations, using 1:10,000 dilution of libraries and size-correction with an average fragment size of 300 nt (ROI 1) and 285 nt (ROI 2). qPCR was performed on the Applied Biosystems ViiA 7 instrument and values were used to pool individual libraries at equimolar ratios to target ~5 M read pairs per library. Final library pools were size-selected using PAGE purification on a 15% TBE gel followed by a crush-and-soak method (Sambrook and Russell, 2006), or with BluePippin 2% gel to select fragments from 250-340 bp. Following size-selection, final pools were quantified by High Sensitivity DNA Kit prior to sequencing.

### Next-generation sequencing

All libraries were sequenced 2 × 150 bp (paired-end) on Illumina platforms with minimum 5% PhiX. Libraries were sequenced at MedGenome, Inc. on the HiSeq X or at Novogene Corporation, Inc. on the NovaSeq 6000. Raw FASTQs were obtained directly from the vendor.

### Ribosome profiling analysis

Analysis was conducted with RiboFlow (umi_devel branch) and RiboR (Ozadam et al, 2020). The reference files provided with RiboFlow were modified to include a Flag tag upstream of the COMT coding sequence to enable mapping of footprints arising from both endogenous COMT and the stably-expressed COMT transgene (ENST00000361682.11 modified with Flag tag). Footprints with lengths between 28 and 35 (inclusive) were used for analysis. A-, P-, and E-site offsets from the 5′ end of the footprint were determined by examining the metagene plots centered at the stop codon. The A-site offset was determined to be 16 nt. This value was used to nominate positions with ribosomal pausing in Figs. 2F and 4C. Positions of ribosome pausing were defined as those 5′ RPF ends that exceeded the 90th percentile in counts per million across the coding region.

### Variant calling and preprocessing

Variants were called with satmut_utils v1.0.3-dev001 (Hoskins, 2023). Analysis utilized the satmut_utils *call* parameters "-n 2 -m 1

--r1_fiveprime_adapters `TACACGACGCTCTTCCGATCT` --r1_threeprime_adapters `AGATCGGAAGAGCACACGCT` --r2_fiveprime_adapters `AGACGTGTGCTCTTCCGATCT` --r2_threeprime_adapters `AGATCGGAAGAGCGTCGTGTA`."

Random forest models were trained on simulated data as previously described for libraries prepared from the amplicon method (Hoskins et al, 2023). Variants were post-processed by the following criteria: variants are within the mutagenesis ROIs; 2) nonsense variants must match the amber codon UAG; 3) $\log_{10}$ variant frequency is $>-5.8$; 4) SNP variants with false positive random forest predictions ($p > 0.5$) in more than half of the gDNA libraries were filtered out. Amino acid positions were adjusted to account for the Flag tag such that positions align with the endogenous membrane-COMT protein isoform.

Frequencies were normalized to the wild-type count at each position (variant count $+ 0.5$/wild-type count $+ 0.5$) (Rubin et al, 2017). Variants in each input source with normalized frequencies lower than the negative control (wild-type plasmid template) were filtered out. Normalized frequencies were batch-corrected to account for the library preparation experiment using ComBat (Leek et al, 2022) with default parameters. Total gDNA and total RNA frequencies were adjusted with a model matrix, whereas no model matrix was used for polysomal RNA and flow cytometry gDNA libraries, as readouts to be compared were in different batches. The median among technical replicates was computed for each biological replicate (independent stable cell line). For correlation between readouts and multi-layer unsupervised learning, the median frequency among biological replicates was first computed.

### Analysis of COMT variant differential abundance between gene expression layers

We only considered variants that were observed in all biological replicates for each readout to be compared such that median natural log-frequency after wild-type normalization and batch correction was greater than $-10$. These frequencies were converted to counts per million and input to ALDEx2 (Fernandes et al, 2013, 2014; Gloor et al, 2016; Gloor, 2021) functions aldex.clr, aldex.effect, and aldex.ttest with parameters "paired=FALSE, denom="iqlr", mc.samples=128".

ALDEx2 effect size is an estimate of the median standardized difference between groups. To avoid any distributional assumptions for standardization, ALDEx2 uses a permutation based non-parametric estimate of dispersion. We opted to use ALDEx2 75% confidence intervals instead of reporting standard deviations on fold changes. The confidence intervals are determined by Monte Carlo methods that produce a posterior probability distribution of the observed data given repeated sampling. The comparisons made were (1) total RNA to total gDNA; (2) polysome metafraction 3 or 4 to total RNA; (3) flow cytometry population 1 and 3 to total gDNA. Polysome metafraction 1 and 2 and flow cytometry population 4 were not used for analysis due to low yield and sequencing depth. To estimate the fold change for nonsense variant effects on RNA abundance, limma-trend (Ritchie et al, 2015) was used on variant log frequencies using empirical Bayes moderation with an estimated proportion of differential genes of 0.03. The point estimate for nonsense variant effects on RNA abundance was derived from nonsense variants detected in all biological replicates.

Transcription factor binding site analysis. For variants with significant effects on RNA abundance (FDR < 0.1), transcription factor binding sites (TFBSs) were searched in the wild-type template sequence and variant sequences using TFBSTools::searchSeq (Tan and Lenhard, 2016) and vertebrate, non-redundant probability frequency matrices (PFMs) from the JASPAR database (Rauluseviciute et al, 2023). To ensure motifs could be properly scored for variants near the termini of the ROIs, sequences were buffered by 33 nt template sequence on each side. Only transcription factor PFMs with uppercase gene names (human genes) were used for scoring. Sequences were scored for all TFBSs irrespective of motif score (searchSeq min.score=0). TFBSs with identical scores for the wild-type template and variant sequences were filtered out (TFBS is not altered by the variant). Top candidate TFBSs generated or abrogated by a variant were selected by the following criteria: (1) For variants generating a TFBS, the relative score of the variant TFBS is $\geq 0.8$, and for variants abrogating a TFBS in the template, the relative score of the template TFBS is $\geq 0.8$. (2) The TFBS with the greatest difference between variant and template scores was selected and reported for each variant in Dataset EV1.

### RNA secondary structure analysis

The RNAfold web server (Sabarinathan et al, 2013b; Gruber et al, 2008) was used to generate RNA structure plots in Figs. 1–2. In Fig. 1A, exons 2–3 (NM_000754.4), including the Flag tag, were folded using default parameters. In Fig. 2G, a 120 nt sequence spanning the H62-V63 codons was folded. Minimum free energy structures were plotted. To predict variant impacts on RNA structure, command-line RNAsnp (Sabarinathan et al, 2013a) was used with default parameters (global folding mode 1, 200 nt window flanking the variant). Except for Fig. 5G, variants with a max distance $p$ value <0.1 (measured by max Euclidean distance from the wild-type base pairing probabilities) were reported. RNA probing data from the RASP database (Li et al, 2021) was examined with the Integrative Genome Viewer.

### Analysis of COMT variant effects on protein abundance

We identified variants enriched in flow cytometry population 3 (P3) compared to total gDNA using ALDEx2. A variant enriched in P3 (FDR < 0.05) is interpreted as one that lowers protein abundance. In Fig. 5, effect sizes were multiplied by $-1$ to indicate a negative effect. PyMol Open-Source (DeLano, 2002) was used to visualize the S-COMT structure (PDB 3BWM). To keep variant annotations consistent in the text, we mapped annotations relative to MB-COMT onto the S-COMT crystal structure.

### Codon stability and ribosome pausing analysis

Codon stability coefficients and ribosome dwell time scores were obtained (Narula et al, 2019; Wu et al, 2019), and the difference in these scores between the alternate and reference codon was computed and compared to the direction of effect. For codon stability coefficient data, one-sided Wilcoxon rank sum tests were used with the expectation that variants that decrease ribosome load have lower codon stability coefficient differences. All other data was tested with two-sided Wilcoxon rank sum tests.

### Unsupervised machine learning of multi-layer gene expression

Median frequencies across biological replicates for each gene expression layer were first transformed to interquartile log ratios (IQLRs). To compute IQLRs, readouts were reclosed (divide each

variant frequency by the total sum of frequencies); the frequency matrix was converted to centered log ratios (CLRs); the variance in CLRs across readouts was computed for each variant and the those with a variance within the interquartile range were selected (interquartile variable features); finally, the log ratio between each variant frequency and the geometric mean of the interquartile variable features was taken. IQLRs were scaled (centered and standardized within a layer), then for each variant, robust centering to the median was performed across layers. This ensures the nominal variant abundance in the library is not learned. Using the Kohonen R package (Wehrens and Buydens, 2007), a self-organizing map (SOM) (Kohonen, 1982) with 260 neurons was trained using a $13 \times 20$ toroidal grid with hexagonal topology and bubble neighborhood. The optimal initialization was determined by training 500 SOMs with different random initialization seeds, and choosing the seed that minimizes the distance between the input data and the matching codebook vectors. The dataset was presented to the network 100 times with a linearly decreasing learning rate of 0.05. Subsequently, neurons were clustered with affinity propagation clustering (Frey and Dueck, 2007) using the R apcluster package (Bodenhofer et al, 2011). The pairwise similarity matrix used negative squared distances, and exemplar preferences were initialized with a $q$ parameter value of 0.1.

## Data availability

The datasets and computer code produced in this study are available in the following databases: DNA-Seq and RNA-Seq data: Gene Expression Omnibus GSE246139; modeling computer scripts: GitHub (https://github.com/ijhoskins/integrated_MAVEs); processed data: MAVEdb urn:mavedb: 00000661; raw data: Zenodo (https://doi.org/10.5281/zenodo.10403931).

## Peer review information

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

## Acknowledgements

This work was supported in part by NIH grant R35GM150667 and the Welch Foundation grant F-2027-20230405 to CC. CC is a CPRIT Scholar in Cancer Research supported by CPRIT grant RR180042. We would like to thank the Doug Fowler lab for providing the HEK293T landing pad cell line and the Fritz Roth lab for providing the plasmid backbone used for integration into the landing pad. We also thank the Center for Biomedical Research Support Microscopy and Imaging Facility at UT Austin (RRID# SCR_021756) for instrumentation that supported this work.

## Author contributions

**Ian Hoskins**: Formal analysis; Validation; Investigation; Visualization; Methodology; Writing—original draft. **Shilpa Rao**: Investigation. **Charisma Tante**: Investigation. **Can Cenik**: Conceptualization; Supervision; Funding acquisition; Project administration; Writing—review and editing.

## Disclosure and competing interests statement

The authors declare no competing interests.

# Expanded View Figures

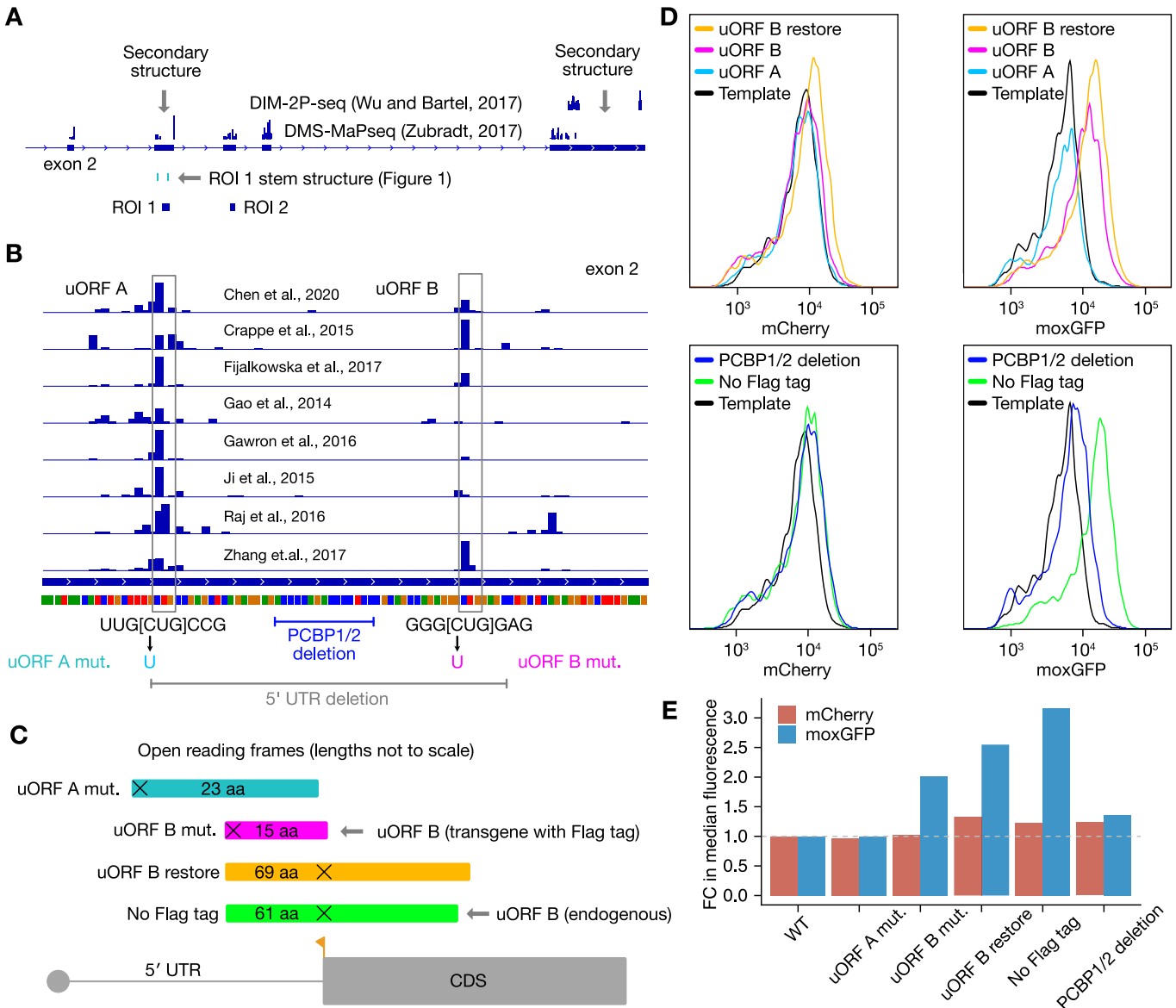

**Figure EV1.** *COMT* 3′ UTR RNA secondary structure, uncharacterized uORFs, and noncoding variant effects on mRNA and protein abundance.

(**A**) *COMT* RNA structure in the 3′ UTR. RNA probing data for HEK293T cells, available from the RASP database (Li et al, 2021), is shown. DIM-2P-seq employs polyA capture to enrich for structures at the 3′ end (Wu and Bartel, 2017). Gray arrows indicate sites of potential secondary structure. (**B**) *COMT* uORF translation initiation sites. TIS-ribosome profiling data from GWIPS-viz (Michel et al, 2014) was visualized. Two uORFs starting at CUG were identified in the 5′ untranslated exon 2 (NM_000754.4) in multiple cell lines including HEK293T. At bottom, isolated mutations and deletions are annotated, which are assayed in (**D**) and Fig. EV2 (5′ UTR del). (**C**) Schematic of uORF A and B reading frames. Black crosses indicate locations of the mutants. (**D**) Flow cytometry data of noncoding and Flag tag variants. See (**B, C**) for schematics. "uORF B restore" and "No Flag tag" variants restore a uORF B frame that overlaps the canonical ORF. "PCBP1/2 deletion" indicates deletion of poly(rC) binding protein (PCBP1, PCBP2) motifs. All variants with a reported fold change were significant by two-sided Wilcoxon rank sum tests at $p < 1 \times 10^{-9}$. (**E**) Fold change in median fluorescence for variants in (**D**). TIS translation initiation site, DMS dimethyl sulfate mutation profiling methods, uORF upstream open reading frame, UTR untranslated region.

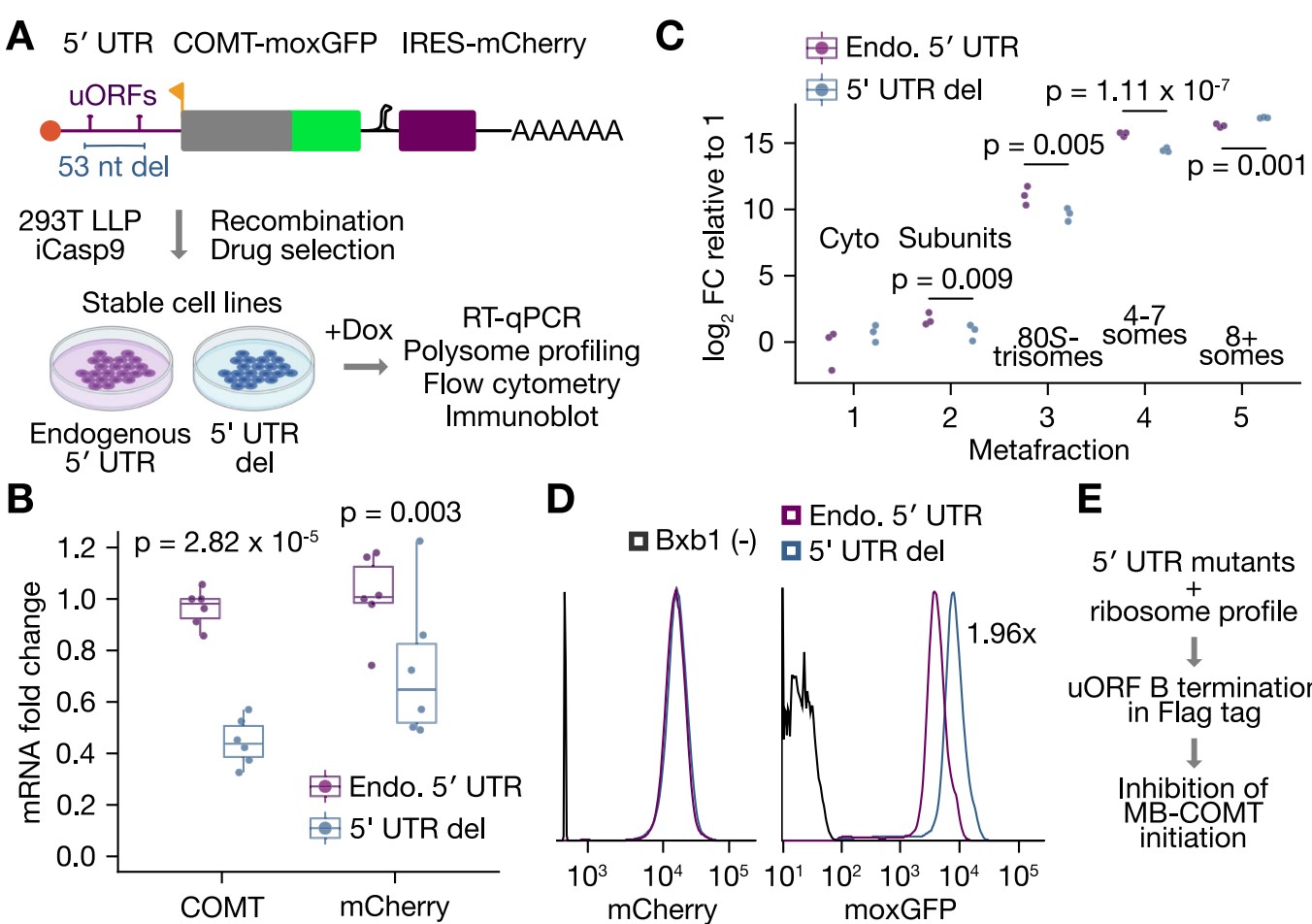

**Figure EV2. The *COMT* 5′ UTR regulates mRNA and protein abundance.**

In (B, C), significance was determined by two-sided Welch *t* tests. (A) Schematic of *COMT* transgene and strategy for stable expression. A *COMT* transgene with a partial 5′ UTR (exon 2 only) and a 53 nt deletion was cloned with a N-terminal Flag, C-terminal moxGFP, and bicistronic mCherry ORF. Transgenes were stably expressed in a HEK293T landing pad line and assayed following induction of transcription with Doxycycline (Dox). (B) The 5′ UTR del decreases *COMT* mRNA abundance. The mRNA fold change was determined by RT-qPCR and plotted relative to the median of Endo. 5′ UTR technical replicates ($N = 6$). The boxplot range is from the first to third quartile, center is the median, and whiskers extend to the largest and smallest values no greater than 1.5 * interquartile range. (C) The 5′ UTR del mRNA is decreased in low polysomes and enriched in high polysomes. RNA from raw polysome fractions were pooled into "metafractions" and RNA was extracted and assayed by RT-qPCR ($N = 3$ technical replicates). (D) The 5′ UTR del leads to higher COMT protein abundance. Flow cytometry results of the transgenic cell lines and a non-recombined (negative control) cell line are shown. (E) Proposed model for regulation by the 5′ UTR. 5′ UTR mutants indicated initiation at uORF B and termination within the Flag tag blocks initiation at the canonical TIS. FC fold change, TIS translation initiation site, moxGFP monomeric, oxidation-resistant GFP (Costantini et al, 2015), IRES internal ribosome entry site, 293T LLP iCasp9 lentiviral landing pad (Matreyek et al, 2020).

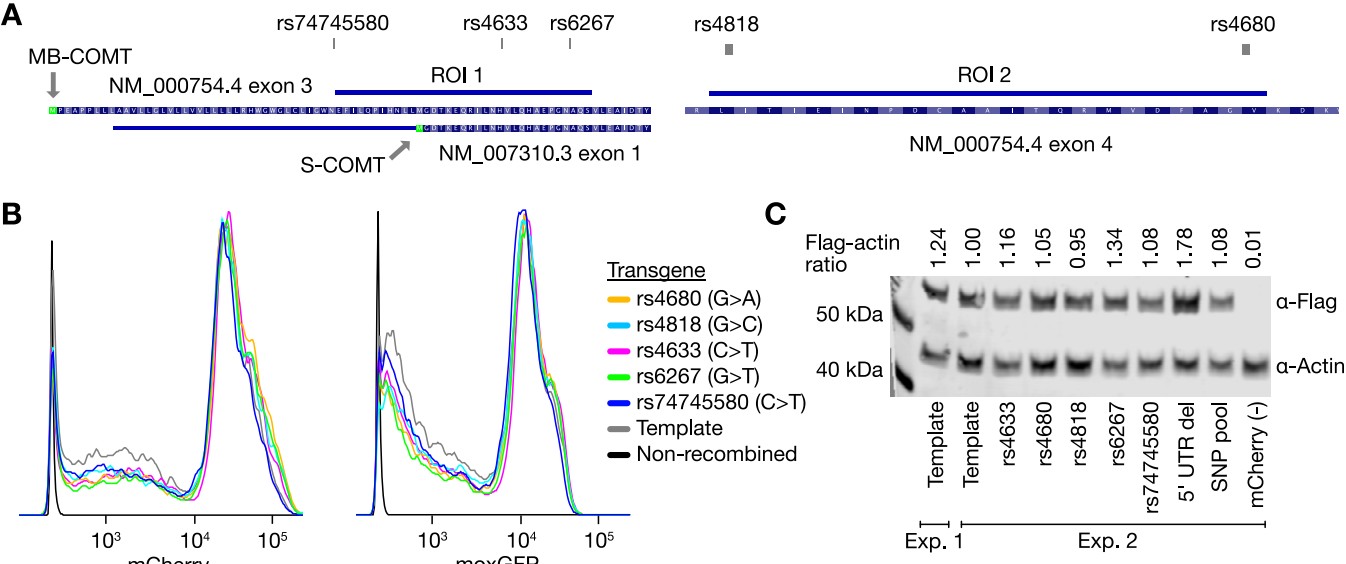

**Figure EV3.   *COMT* common population SNPs have no impact on mRNA and protein abundance in the absence of complete UTRs.**

(A) *COMT* regions of interest and location of common population SNPs. (B) No impact of common population variants on *COMT* mRNA or protein abundance. Fluorescence of mCherry (mRNA proxy) and moxGFP (protein) is plotted for each common population variant (indicated with dbSNP identifiers), along with a non-recombined negative control and the non-mutagenized template, which consists of the LPS haplotype (Nackley et al, 2006) without the intronic SNP rs6269. (C) Immunoblotting of common population SNPs. Background-corrected Flag signal to Actin signal from densitometry analysis is indicated at the top of each lane. For the template transgene, lysates from two separate experiments were analyzed. "5′ UTR del" is the deletion mutant analyzed in Fig. EV2. "SNP pool" refers to a stable cell line with all common population variants. "mCherry (−)" refers to a negative control for Flag expression, an untagged mCherry payload.

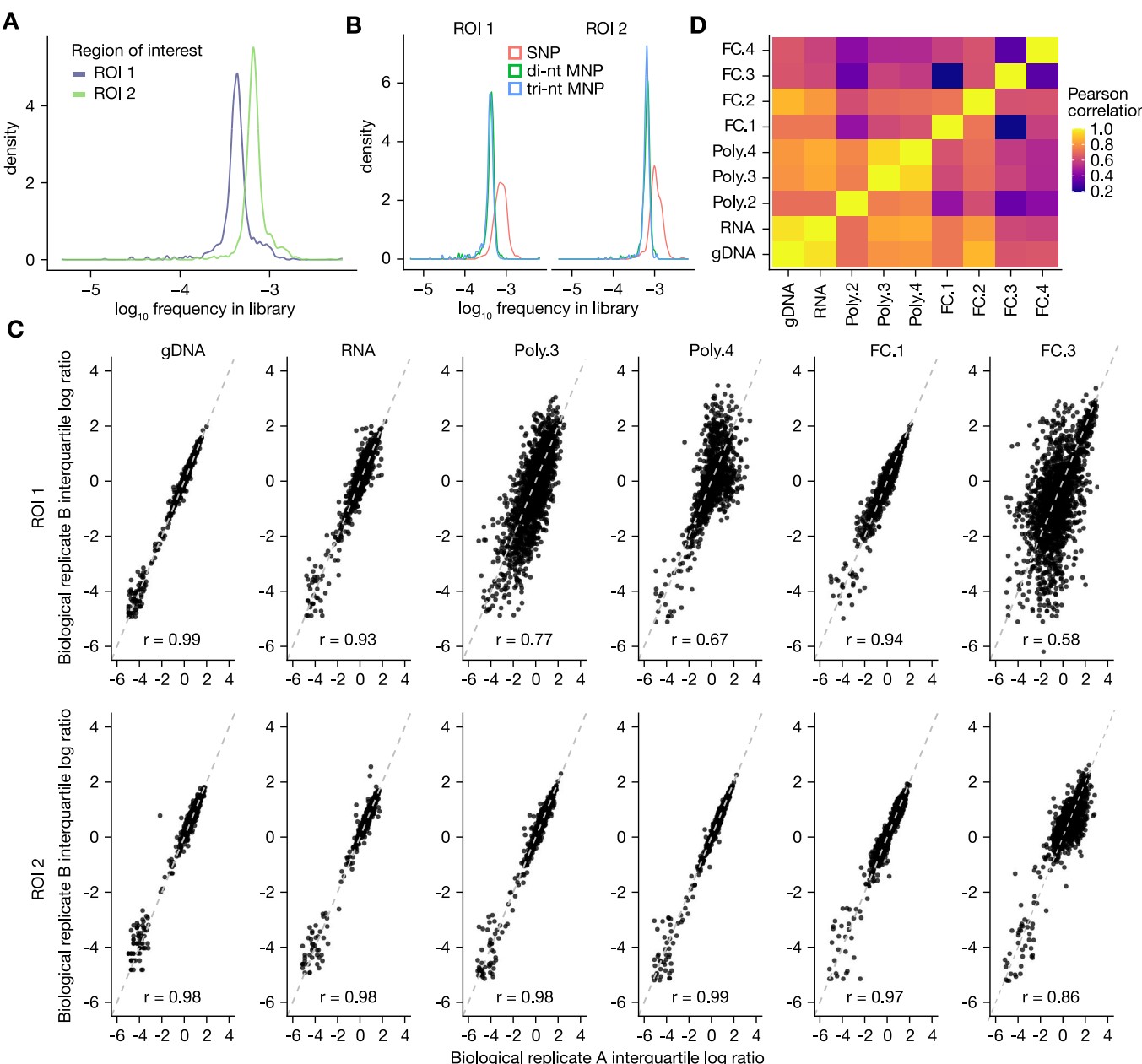

**Figure EV4.  *COMT* library quality control and reproducibility of variant abundance estimates.**

In (A, B), plotted is kernel density estimation of $\log_{10}$ transformed frequencies reported in the Twist Biosciences quality control report. In panels C and D, and FC.1-4 refer to the four sorted flow cytometry populations, and Poly.2-4 refer to the second–fourth metafractions (see Fig. EV5D for schematic). (A) Variant uniformity for *COMT* target ROIs. (B) Frequency distributions by variant type. SNPs are at higher frequencies than di-nt and tri-nt MNPs, which have two and three mismatches, respectively. (C) Biological replicate Pearson correlation. Two of the four biological replicates are shown as examples. Metafractions and flow cytometry populations that were not subsequently analyzed (Poly.1, Poly.2, FC.2, FC.4) are omitted. (D) Correlation between gene expression readouts. Plotted is Pearson correlation of the median $\log_{10}$ frequency of biological replicates between readouts.

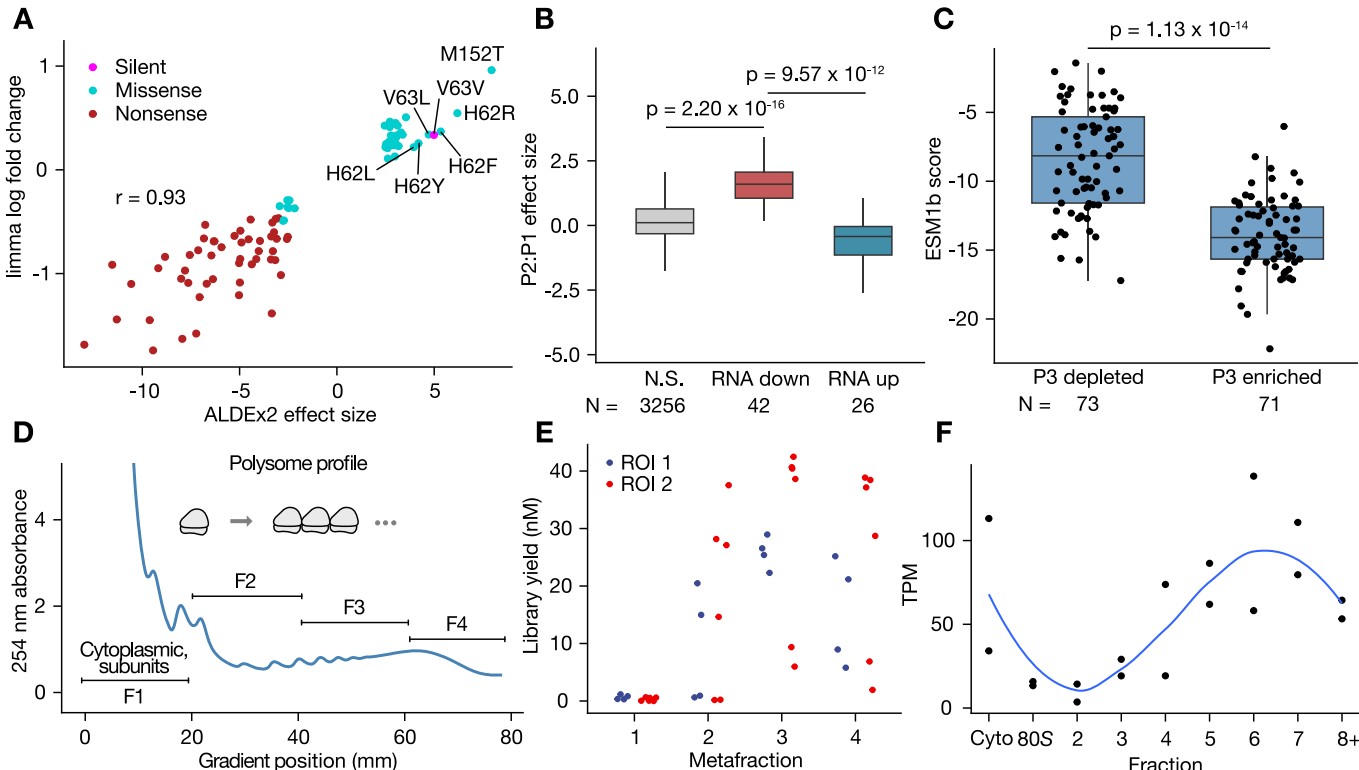

**Figure EV5. Variant effect size measures, correlation of protein abundance effects, and low versus high polysome fractions.**

In (**B**, **C**), significance was determined with two-sided Wilcoxon rank sum tests. The boxplot range is from the first to third quartile, center is the median, and whiskers extend to the largest and smallest values no greater than 1.5 * interquartile range. (**A**) Relation between limma and ALDEx2 effect measurements for RNA abundance. Pearson correlation coefficient is shown for variants at a FDR < 0.1. (**B**) Variants decreasing RNA abundance are enriched in flow cytometry P2 compared to P1. Outliers >1.5 * interquartile range were omitted for clarity. (**C**) Variants enriched in P3 have more negative ESM1b (Brandes et al, 2023) scores (log likelihood ratios) than those depleted from P3. (**D**) Representative polysome trace indicating the fraction pooling strategy. F1 and F2 were not subsequently analyzed due to inadequate library yield or reproducibility. (**E**) Yields for polysome metafraction libraries. Yield was quantified by qPCR using Illumina adapter specific primers ("Methods"). (**F**) Polysome profile of endogenous *COMT* in HEK293T cells. Transcripts per million mapped reads (TPM) is plotted using gene-level data from TriP-seq (Floor and Doudna, 2016). Blue line is a loess fit with confidence intervals omitted for clarity. Cyto cytoplasmic fraction.

