## [Peer Review File · Molecular Systems Biology]

Integrated multiplexed assays of variant effect reveal determinants of catechol-O-methyltransferase gene expression

Ian Hoskins, Shilpa Rao, Charisma Tante, and Can Cenik

Corresponding author(s): Can Cenik (ccenik@austin.utexas.edu)

Review Timeline:

Transfer from Review Commons:	1st Nov 23
Editorial Decision:	27th Nov 23
Revision Received:	21st Dec 23
Editorial Decision:	3rd Jan 24
Revision Received:	16th Jan 24
Accepted:	18th Jan 24

Editor: Maria Polychronidou

Transaction Report: This manuscript was transferred to Molecular Systems Biology following peer review at Review Commons.

Review #1

1. Evidence, reproducibility and clarity:

Evidence, reproducibility and clarity (Required)

In this manuscript, Hoskins et al describe analyses of the effects of sequence variation on RNA levels, protein levels, and ribosome loading for the COMT gene. They use multiple experimental approaches to assay these levels and report on how sequence differences affect expression. Overall, the paper is interesting in that it presents a very deep dive into the effects of sequence variation on gene expression, including in coding sequences. However, there are some issues with the polysome loading assay technique and there are substantial issues with the figure presentation, which is often confusing.

****Major comments:****

1. Figures:

Fig 1C needs a cartoon description to show where the UTRs are. Y-axis should say "Ribo-seq CPM"

Sup Fig 1A confusing, what is "start" what is the point of this panel?

Sup Fig 1B what is PCBP del?

Sup Fig 1C what is "uORF B restore"? The description in the figure legend is not interpretable. Draw diagrams of the mutations that tell the reader what was assayed and why it was assayed. Why are there multiplication factors listed (e.g. 1.33X)? The data are depicted on a log scale, which makes it difficult to appreciate the fold-effects of the mutations (e.g. does uORFA mutation increase expression 1.5-fold?). Please calculate median expression values and report them on a bar graph or something like that so readers can interpret the results.

Fig 2A. It's hard to understand the cartoon diagram of the expression reporter construct. Why is +Dox shown here? Does that induce transcription?

Fig 2B. What's on the x-axis? is it $\text{Log}_2(\text{RNA/gDNA})$ from sequencing? is it Log_2 or Log_{10} or Ln ?

Fig 2C. What's on the y-axis (same question). I think it's $\text{Log}_X(\text{mutant/wt})\text{RNA level}$?

Fig 2D. What's on the y-axis now? Fold-difference (not log transformed)?

Fig 2E. The scale bar is flipped vs. normal convention. This is also log transformed, but it's not labeled. Please label as log(whatever) and put the negative values on the left side of the bar (red on the left, blue on the right).

Fig 2F y-axis should say Ribo-seq CPM.

Fig 3A - please separate the graphs more. Did you sort cells from ROI2 into populations, or just cells from ROI1?

Fig3C-F What's the "effect size" mean on these graphs?

Fig3D It looks like the colors have switched for positive / negative "effects" on the heat map compared to Figure 2E. Please define what "median effect" means and be consistent with comparison to figure 2E.

Figure 4 what does effect size mean, what's the log-transformed scale (log2, 10, etc) same issues from earlier figures.

Figure 5 "effect size"

2. "Codon stability" should always be "Codon Stability Coefficient", maybe use "CSC". Otherwise it's confusing.

3. Flow cytometry section talks about "RNA fluorescence", which is confusing. You need to explain that it's IRES-driven mCherry as a proxy for the level of RNA first. It would also help to state explicitly that you sorted the cells into four populations, and define them all first before describing the results.

4. What are DeMask scores? How are they related to conservation or amino acid properties? If you define these, you can help the reader interpret the result.

5. There are several issues with the Polysome gradient fractionation. The gradients did not separate 40S, 60S, and monosomal fractions, so it's hard to tell how many ribosomes correspond to each peak on the gradient graph in Figure S5. This is probably because the authors used a 20-50% gradient instead of a lower percentage on top. More significantly, variations in the coding region of COMT are likely affecting the polysome association in ways the authors didn't consider. Nonsense codons will simply make the orf a lot shorter, hence fewer ribosomes. This may have nothing to do with NMD. Silent and missense variants may have unpredictable effects because they may make translation faster (fewer ribosomes) or slower (more ribosomes) on the reporter. This could lead to more ribosomes with less protein or fewer ribosomes with more protein. The reporter RNA also has an IRES loading mCherry on it, which

probably helps blunt or dampen the effects of the COMT sequence variants on polysome location distribution. Overall, the design of the polysome assay is probably very limited in power to detect changes in ribosome loading (four fractions, limited separation by 20-50 gradient, IRES loading, etc). This is partially addressed in the limitations section, but these issues could be discussed in more detail.

****Referees cross-commenting****

I generally agree with the other reviews. Reviewer 2 asks a lot of clarifying questions, which is in line with my comments and suggestions to clarify the presentation of results in the figures. Reviewer 3 has some similar comments and also asks for validation of a few of the MPRA results, which I agree would strengthen the manuscript.

2. Significance:

Significance (Required)

The study is novel in that it assays both 5' UTR and a wide range of protein coding sequence variants for effects on RNA and protein levels from a clinically important gene, COMT. The manuscript reports that most protein coding variants have modest effects on RNA levels, and that the minority of variants that do affect RNA levels are not predictable due to their affect on codon usage. The work also determines the distribution of effects of variants on protein levels, finding a variety of effects on expression. Interestingly, the authors found SNPs that affect ribosome loading generally affect RNA structure of the COMT coding region, rather than affecting codon usage.

This should appeal to many different communities of biologists - gene expression experts, geneticists, and clinical neurobiologists who focus on COMT. So there is a potential for fairly broad interest. The main limitations to the work are in a lack of clarity in the figures and perhaps in the underdeveloped nature of the discussion section. The discussion section reports new results (SNP associations that affect expression). These would make more sense in the results section, such that the discussion could do a better job relating the impact of sequence variants on expression levels to prior work to highlight the novelty.

3. How much time do you estimate the authors will need to complete the suggested revisions:

Estimated time to Complete Revisions (Required)

(Decision Recommendation)

Between 1 and 3 months

Yes

Review #2

1. Evidence, reproducibility and clarity:

Evidence, reproducibility and clarity (Required)

****Summary:****

Hoskins and colleagues expressed a reporter containing all silent, missense, and nonsense codons at 58 amino acid positions in the human COMT gene in HEK293T cells and measured levels of DNA, bulk RNA, and pooled polysomal mRNA. They included a C-terminal translational GFP fusion and a downstream transcriptional mCherry fusion in the reporter in order to also bin variants by their relative protein and mRNA levels by flow cytometry. They hypothesized that RNA structure, in-part by mediating uORF translation, influences COMT gene expression. The authors conclude by identifying previously-uncharacterized COMT variants that, in this reporter system, affect RNA abundance and ribosome load.

We generally found the results of this paper convincing and clear. We do not have major comments, but have many minor comments that we hope the authors can address. These comments mostly deal with clarification on analysis metrics and giving

recommendations on data presentation.

****Minor comments:****

In Figure 2C, the vertical axis reads "Median between-group difference". How was this metric calculated and normalized? We also agree that nonsense mutations having consistently-detrimental effects on RNA abundance is reassuring, but recommend more explanation as to why the difference in the effects of silence and missense mutations between regions may be biologically relevant.

In Figure 3, we believe that the authors are claiming that lower RNA abundance causes lower protein abundance in some variants. However, this data only reports on protein abundance relative to transcript abundance, not absolute protein abundance. We think the claim should be revised to (1) clarify that the authors are measuring protein per mRNA, and (2) express that lower mRNA amounts are more likely to co-occur with lower protein amounts, but that this data does not support any causative model.

On page 9, the authors claim that their data supports a model that rs4633 increases RNA abundance, leading to higher COMT expression. Can the authors rule out a model whereby rs4633 facilitates translation initiation, as suggested by Tsao et al. 2011, leading to both an increase in mRNA and protein abundance?

The paper references "effect size" at multiple points (e.g. "polysome effect size") but we could not find this term explicitly defined (for example: for the polysome effect size, were RNA counts for each polysome fraction divided by the relative abundance of that RNA in total RNA?)

Could you elaborate on how you define "protein abundance and "effect size: in Figure 5G? How is enrichment in P3 or P1 calculated?

Were 3396 variants considered for all readouts in this paper? How many of these variants were present in each ROI? It may be worth clarifying sample sizes.

How did Twist generate these mutagenized sequences? We assumed that they used error-prone PCR due to the mention of multiple nucleotide polymorphisms, but couldn't find an explicit answer.

In the methods, it may be worth elaborating on the composition of the HsCD00617865 plasmid. For example: this COMT reporter is under the control of a constitutively-expressed T7 promoter, correct?

In Supplementary Figures 4 and 5, it would be helpful to explicitly say that you are reporting Pearson correlations between biological replicates.

"After summarizing biological replicates (N=4) for each readout...": how did the authors summarize biological replicates? Were counts averaged?

The authors used pairwise correlations between flow cytometry fractions, polysome fractions, and total RNA/gDNA as indications of data quality. Do the authors expect for these counts to be strongly correlated? We would not necessarily expect to see a strong correlation between ribosome load and RNA/gDNA.

The authors may need to check that their standard deviations on fold changes are properly reported. We would expect standard deviation bounds to be symmetric for log fold changes, but not on unlogged fold changes - for example see page 8, for the sentence "our point estimate for nonsense variant effects on COMT RNA abundance was approximately a two-fold decrease relative to the gDNA frequency (fold change of 0.43 +/- 0.13; mean +/- standard deviation; Methods)."

On page 10, the authors say that their data suggests that hydrophobicity in the early coding region of COMT may be important for COMT folding. If this is the case, would we expect to see this effect in flow cytometry data (which is affected by protein degradation) and not polysome profiling (which is unaffected by post-translational protein degradation)?

We believe that we would have some trouble replicating the analysis from this paper from the raw data, given that the bulk of the analysis on GitHub is presented as a single R Markdown file, with references to local files to which we do not have access. We recommend that the authors add additional documentation to their repository to facilitate re-analysis.

In Figure 1B, indicating that more signal indicates less structure (in the legend or the figure itself) may assist readers who are unfamiliar with DMS-seq.

Figure 1C does a great job presenting evidence for the translation of uORFs, but does not seem to flow with the overall argument of the paper, so may fit better in the supplement.

We believe there is a typo in the Figure 1 legend that should read "K562" instead of "H562".

You also gated to separate into P1-P4, correct? Can you also show the bounds of that gating strategy in Figure 3A?

We find Figure 3F very compelling. Do you have any theories as to why mutating I59-H66 to nonpolar, uncharged residues leads to increased COMT expression? There appears to be a non-negligible proportion of di- and tri- nucleotide polymorphisms in Supplementary Figure 4. Were these excluded in downstream analyses?

A minor typo in the discussion reads "fluoresce".

2. Significance:

Significance (Required)

Describe the nature and significance of the advance (e.g. conceptual, technical, clinical) for the field.

This work investigated the regulatory effects of thousands of coding variants in the COMT gene, focusing on two regions with clinical significance, by using high-throughput reporter assays. The results from this will be useful for clinical scientists interested in understanding the impacts of COMT mutations and be a useful framework for other systems/computational biologists to understand the impacts of coding mutations across different levels of regulatory function. Mutations in protein regions, if having a function, are classically known to interfere with protein function. There are fewer large-scale efforts to understand the impacts of coding mutations affecting expression through potentially changing of RNA structure or codon optimization - this work has contributed towards that frontier.

Place the work in the context of the existing literature (provide references, where appropriate).

This is (as far as I am aware) the first paper that has integrated high-throughput screens massively parallel reporter assays from RNA degradation, ribosomal load, and flow cytometry. Previous papers have tended to measure on expression regulation on only one dimension (i.e. Greisemer et al. 2023 on RNA degradation, Sample et al. 2019 on ribosomal load, and de Boer et al. 2020 on protein expression).

State what audience might be interested in and influenced by the reported findings.

Clinicians/researchers interested in COMT, computational biologists, geneticists and

potentially structural biologists interested in understanding the consequences of amino acid mutations on RNA/protein expression

Define your field of expertise with a few keywords to help the authors contextualize your point of view. Indicate if there are any parts of the paper that you do not have sufficient expertise to evaluate.

Genomics, Massively parallel reporter assays, High-throughput regulatory screens.

3. How much time do you estimate the authors will need to complete the suggested revisions:

Estimated time to Complete Revisions (Required)

(Decision Recommendation)

Between 1 and 3 months

Yes

Review #3

1. Evidence, reproducibility and clarity:

Evidence, reproducibility and clarity (Required)

This manuscript reports on transcript sequence variants that affect expression of the gene COMT. Targeted analysis of SNPs identifies 5' UTR variants that affect COMT, leading to the identification of translated uORFs. Common coding sequence SNPs do not affect COMT expression, however. Massively parallel analyses of mRNA abundance, protein abundance, and translation are combined to look more broadly at coding sequence variants. These analyses focus on regions of predicted structure in the COMT transcript. Both silent and missense mutations that increase mRNA abundance are identified. Protein abundance is then measured and many missense mutations are found to change protein levels. To address translation directly, analysis of polysome loading is performed and significant differences are identified, although technical challenges limit data quality in these experiments. These different experiments are then analyzed jointly to classify mutation effects and identify a class of silent mutations with expression effects, leading to a proposal that these act through structure.

The joint, integrative analysis of COMT variants through a range of methods allows clearer insights into interconnected post-transcriptional effects. The massively parallel experiments generate high-quality data, although targeted validation of key results would strengthen the work. The findings advance our understanding of silent variant effects, which remains an open question, and technical innovations could find broader applications.

I do have concerns with the present version of this work.

1. There is no validation presented for high-throughput experimental data. I would say that validating the effects of M152T and V63V variants from Figure 2B would substantially strengthen the work and support key conclusions.
2. In the fluorescent reporter scheme, it seems that variants reducing mRNA abundance should be enriched in the "P2" gate region relative to "P1", as they would have lower mRNA abundance and correspondingly lower protein abundance. However, this analysis is not performed, and instead P1 and P3 are compared (Figure 3G), which would seem to focus on protein-level effects.
3. In general the work classifies variants in several different ways and it would help to be a little clearer in naming these classes. For instance, in describing the FACS-based analysis of variant expression it is written, "protein fluorescence conditioned on RNA fluorescence" which is confusing at best-it's a fluorescence-based measurement that is used indirectly to measure COMT reporter abundance.
4. Likewise, the populations with shifted GFP/mCherry ratio in this assay are described as "uncorrelated" populations, which is opaque and somewhat inaccurate-there seems to be a correlation in this group but at a different ratio.
5. In the same way, "deleterious variants" is used to describe protein abundance

changes, but this term implies a fitness effect and is not very specific.

6. In discussing the effects of missense COMT variants on protein levels, there is an implicit assumption that degradation of mis-folded protein (or perhaps properly-folded protein with excess hydrophobic exposure?) explains these effects. This is plausible, but it would help to lay out this reasoning more clearly.

7. It is written that, "In line with codon stability as a predictor of translational efficiency (Presnyak et al., 2015), variants with low codon optimality were depleted from polysomes compared to variants with optimal codons". However, this mis-states the conclusions of the cited study, which notes, "Importantly, under normal conditions the ribosome occupancy of the HIS3 opt and non-opt constructs was determined to be similar (Fig. 6B)".

8. It is written that, "One intriguing possibility is to develop multiplexed assays of variant effect on RNA folding, using mutational profiling RNA probing methods (Weng et al., 2020; Zubradt et al., 2017)." How would this differ from the "Mutate and Map" approach in doi://10.1038/nchem.1176 and subsequent work from the same group?

****Referees cross-commenting****

I generally agree with the other reviewers and found that many small points on the figures were confusing, and in some cases the values being computed and displayed were under-specified.

I agree with Reviewer 1 that the polysome fractionation probably has limited power due to experimental design, and that the interpretation of changed ribosome loading is subtle.

2. Significance:

Significance (Required)

The joint, integrative analysis of COMT variants through a range of methods allows clearer insights into interconnected post-transcriptional effects. The massively parallel experiments generate high-quality data, although targeted validation of key results would strengthen the work. The findings advance our understanding of silent variant effects, which remains an open question, and technical innovations could find broader applications.

3. How much time do you estimate the authors will need to complete the suggested revisions:

Estimated time to Complete Revisions (Required)

(Decision Recommendation)

Between 1 and 3 months

Yes

Full Revision

Manuscript number: RC-2023-02121R

Corresponding author(s): Can, Cenik

1. Point-by-point description of the revisions

Reviewer #1

Evidence, reproducibility and clarity

In this manuscript, Hoskins et al describe analyses of the effects of sequence variation on RNA levels, protein levels, and ribosome loading for the COMT gene. They use multiple experimental approaches to assay these levels and report on how sequence differences affect expression. Overall, the paper is interesting in that it presents a very deep dive into the effects of sequence variation on gene expression, including in coding sequences. However, there are some issues with the polysome loading assay technique and there are substantial issues with the figure presentation, which is often confusing.

Response: Thanks for the positive assessment of our manuscript and the constructive feedback regarding the issues with the figure presentation. We have addressed all of these below and they have significantly improved the clarity.

Major comments:

1) Figures:

--Fig 1C needs a cartoon description to show where the UTRs are. Y-axis should say "Ribo-seq CPM"

Response: Fig 1C now includes a schematic and the y-axis is updated. Locations of the uORFs are also now included in Fig 1A.

--Sup Fig 1A confusing, what is "start" what is the point of this panel?

Response: We apologize for the confusing labeling of the panels in Sup Fig 1. "Start" refers to the MB-COMT start codon. We removed this annotation as it is irrelevant to the figure. We included Supplementary Figure 1A to show RNA probing data for the entire transcript. Figure 1A and B only show the regions that encompass the variants assayed in our study.

--Sup Fig 1B what is PCBP del?

Response: "PCBP del" refers to deletion of PCBP1/PCBP2 RNA binding protein motifs. The legend now specifies this.

--Sup Fig 1C what is "uORF B restore"? The description in the figure legend is not interpretable. Draw diagrams of the mutations that tell the reader what was assayed and why it was assayed. Why are there multiplication factors listed (e.g. 1.33X)? The data are depicted on a log scale, which makes it difficult to appreciate the fold-effects of the mutations (e.g. does uORFA mutation increase expression 1.5-fold?). Please calculate median expression values and report them on a bar graph or something like that so readers can interpret the results.

Response: “uORF B restore” refers to restoration of the endogenous uORF B frame with a silent variant in the Flag tag of the transgene. The multiplication factors listed were the fold change in median fluorescence between each mutant and the template (wild-type) transgene. We retained the figures as they show the raw distribution of fluorescence in each cell line, but in response to the reviewer’s suggestion we included a new figure displaying the effects as a bar graph (Supplementary Figure 1E).

--Fig 2A. It's hard to understand the cartoon diagram of the expression reporter construct. Why is +Dox shown here? Does that induce transcription?

Response: The reviewer is correct. “+Dox” indicated addition of Doxycycline to induce transcription before the data collection step. We agree that there may have been too much detail in this diagram and have now removed this for simplicity and indicated this in the Methods section.

--Fig 2B. What's on the x-axis? is it Log2(RNA/gDNA) from sequencing? is it Log2 or Log10 or Ln?

Response: Variant effects in each figure were derived from ALDEx2 analysis, which reports effect size as the median standardized difference between groups. The effect size is not directly interpretable as a log fold change; it takes into account the difference between groups as well as the dispersion. This analysis strategy has been previously demonstrated for analysis of SELEX experiments (Fernandes et al. 2014), which are used to select small populations of cells with specific phenotypes.

ALDEx2 is a robust and principled choice for the analysis of count-compositional datasets, particularly after selection (e.g. sorted cell populations or low-input RNA fractions arising from polysome profiling). While we understand that this choice leads to less easily interpretable effect sizes, the mathematical advantages make ALDEx2 a more appropriate choice for this type of data. In the past, we had used other methods to analyze log frequencies (limma, a frequency based normalization-dependent analysis, as previously employed in Hoskins et al. 2023. Genome Biology) that directly reported fold changes. In our experience, the ALDEx2-derived effect sizes are well-correlated with those estimates (Pearson correlation 0.93 for variants significant at a FDR < 0.1 in the RNA:DNA analysis). We included Supplementary Figure 5A to show this relationship.

--Fig 2C. What's on the y-axis (same question). I think it's LogX(mutant/wt)RNA level?

Response: For consistency with other figures, we replaced Figure 2C to report the effect size statistic as described above.

--Fig 2D. What's on the y-axis now? Fold-difference (not log transformed)?

Response: Please see our response above.

--Fig 2E. The scale bar is flipped vs. normal convention. This is also log transformed, but it's not labeled. Please label as log(whatever) and put the negative values on the left side of the bar (red on the left, blue on the right).

Full Revision

Response: Thanks for the suggestion, we have now updated the scale bar.

--Fig 2F y-axis should say Ribo-seq CPM.

Response: Done

--Fig 3A - please separate the graphs more. Did you sort cells from ROI2 into populations, or just cells from ROI1?

Response: Thanks for the suggestion, we now separate the graphs further. Cells were sorted for both ROI 1 and ROI 2 libraries.

--Fig3C-F What's the "effect size" mean on these graphs?

Response: Please see the response above regarding the effect size estimate from ALDEx2.

--Fig3D It looks like the colors have switched for positive / negative "effects" on the heat map compared to Figure 2E. Please define what "median effect" means and be consistent with comparison to figure 2E.

Response: We intentionally inverted colors for Figure 3. The rationale is that a variant causing low protein abundance corresponds to *enrichment* in P3 compared to gDNA, as opposed to depletion in P3. On the other hand, for effects on RNA abundance and ribosome load, a variant leading to low abundance for these measures is depleted.

--Figure 4 what does effect size mean, what's the log-transformed scale (log2, 10, etc) same issues from earlier figures.

Response: Please see response above.

--Figure 5 "effect size"

Response: The same definition of effect size was used with the exception that effect sizes are multiplied by -1 so that color schemes are consistent for deleterious effects.

2) "Codon stability" should always be "Codon Stability Coefficient", maybe use "CSC". Otherwise it's confusing.

Response: Thanks for the suggestion. This has been updated throughout the manuscript.

3) Flow cytometry section talks about "RNA fluorescence", which is confusing. You need to explain that it's IRES-driven mCherry as a proxy for the level of RNA first. It would also help to state explicitly that you sorted the cells into four populations, and define them all first before describing the results.

Response: We apologize for the use of imprecise language with respect to this reporter. We revised the text to emphasize that mCherry is a proxy for RNA abundance and described the populations first as suggested.

4) What are DeMask scores? How are they related to conservation or amino acid properties? If you define these, you can help the reader interpret the result.

Response: Thanks for the suggestion. We now include a conceptual interpretation of the DeMask score in the relevant section. We also include a comparison to a recent large language model for variant effect prediction (ESM1b, Brandes et al. 2023) which is now reported in Supplementary Figure 5C.

5) There are several issues with the Polysome gradient fractionation. The gradients did not separate 40S, 60S, and monosomal fractions, so it's hard to tell how many ribosomes correspond to each peak on the gradient graph in Figure S5. This is probably because the authors used a 20-50% gradient instead of a lower percentage on top. More significantly, variations in the coding region of COMT are likely affecting the polysome association in ways the authors didn't consider. Nonsense codons will simply make the orf a lot shorter, hence fewer ribosomes. This may have nothing to do with NMD. Silent and missense variants may have unpredictable effects because they may make translation faster (fewer ribosomes) or slower (more ribosomes) on the reporter. This could lead to more ribosomes with less protein or fewer ribosomes with more protein. The reporter RNA also has an IRES loading mCherry on it, which probably helps blunt or dampen the effects of the COMT sequence variants on polysome location distribution. Overall, the design of the polysome assay is probably very limited in power to detect changes in ribosome loading (four fractions, limited separation by 20-50 gradient, IRES loading, etc). This is partially addressed in the limitations section, but these issues could be discussed in more detail.

Response: Given high polysomal association of endogenous *COMT* and our *COMT* transgene (Supplementary Figure 2B, Supplementary Figure 5B-C), we chose a 20-50% sucrose gradient to better resolve changes in ribosome load among heavy polysomes.

We thank the reviewer for offering another valid explanation regarding the depletion of nonsense variants. We have now included a sentence in the discussion to indicate lower ribosome load for nonsense variants may be due to a shorter ORF as opposed to NMD. We further include the potential limitation of the assay due to the presence of the IRES-mCherry.

We agree that variants may have unpredictable effects due to effects on the dynamics of translation elongation. To address this potential limitation, we attempted to devise a selective ribosome profiling strategy by immunoprecipitating N-terminal Flag tagged peptides to enrich ribosomes translating *COMT*. However, we were unable to achieve significant enrichment, limiting our ability to measure variant effects on elongation in a high-throughput manner.

Significance

The study is novel in that it assays both 5' UTR and a wide range of protein coding sequence variants for effects on RNA and protein levels from a clinically important gene, COMT. The manuscript reports that most protein coding variants have modest effects on RNA levels, and that the minority of variants that do affect RNA levels are not predictable due to their affect on codon usage. The work also determines the distribution of effects of variants on protein levels, finding a variety of effects on expression. Interestingly, the authors found SNPs that affect

ribosome loading generally affect RNA structure of the COMT coding region, rather than affecting codon usage.

This should appeal to many different communities of biologists - gene expression experts, geneticists, and clinical neurobiologists who focus on COMT. So there is a potential for fairly broad interest. The main limitations to the work are in a lack of clarity in the figures and perhaps in the underdeveloped nature of the discussion section. The discussion section reports new results (SNP associations that affect expression). These would make more sense in the results section, such that the discussion could do a better job relating the impact of sequence variants on expression levels to prior work to highlight the novelty.

Response: We thank reviewer #1 for their positive assessment of the broad significance of our study. We also thank them for constructive suggestions that led to increased clarity in the presentation. We have moved the analysis of gnomAD variants to the Results section and expanded the discussion.

Reviewer #2

Evidence, reproducibility and clarity

Summary:

Hoskins and colleagues expressed a reporter containing all silent, missense, and nonsense codons at 58 amino acid positions in the human COMT gene in HEK293T cells and measured levels of DNA, bulk RNA, and pooled polysomal mRNA. They included a C-terminal translational GFP fusion and a downstream transcriptional mCherry fusion in the reporter in order to also bin variants by their relative protein and mRNA levels by flow cytometry. They hypothesized that RNA structure, in-part by mediating uORF translation, influences COMT gene expression. The authors conclude by identifying previously-uncharacterized COMT variants that, in this reporter system, affect RNA abundance and ribosome load. We generally found the results of this paper convincing and clear. We do not have major comments, but have many minor comments that we hope the authors can address. These comments mostly deal with clarification on analysis metrics and giving recommendations on data presentation.

Response: Thanks for highlighting the strengths of our study and the constructive suggestions to improve the presentation.

Minor comments:

In Figure 2C, the vertical axis reads "Median between-group difference". How was this metric calculated and normalized? We also agree that nonsense mutations having consistently-detrimental effects on RNA abundance is reassuring, but recommend more explanation as to why the difference in the effects of silence and missense mutations between regions may be biologically relevant.

Response: Variant effects in each figure derive from ALDEx2 analysis, which reports effect size as the median standardized difference between groups. In particular, to avoid any distributional assumptions for standardization, ALDEx2 uses a permutation based non-parametric estimate of dispersion. The effect size is not directly interpretable as a log fold change; it takes into account the difference between groups as well as the max dispersion of the groups. We have now

provided explicit references to the specific R functions that were used to calculate the effect size.

ALDEx2 is robust for analysis of count-compositional datasets, particularly after selection and bottlenecking (e.g. sorted cell populations or low-input RNA fractions arising from polysome profiling). While we have used other methods to analyze log frequencies (limma, a frequency based normalization-dependent analysis, as previously employed in Hoskins et al. 2023. Genome Biology), we opted for the less-interpretable but more robust ALDEx2 analysis to report variant effects between varying nucleic acid inputs.

We currently lack a mechanistic interpretation for the difference in RNA abundance effects between ROI 1 and 2. However, we observed consistent results using a different analysis framework, which makes use of variant frequencies (as in Hoskins et al. 2023 Genome Biology) instead of the centered log ratios used in ALDEx2 analysis, further supporting a biological difference between the two.

In Figure 3, we believe that the authors are claiming that lower RNA abundance causes lower protein abundance in some variants. However, this data only reports on protein abundance relative to transcript abundance, not absolute protein abundance. We think the claim should be revised to (1) clarify that the authors are measuring protein per mRNA, and (2) express that lower mRNA amounts are more likely to co-occur with lower protein amounts, but that this data does not support any causative model.

Response: Thanks for the suggestion. We have now included an explicit description of the experimental design in the results section and noted that we are unable to assign protein abundance effects to underlying RNA abundance effects. In the current setup, we did not sort cells based on the ratio of moxGFP/mCherry fluorescence (protein per mRNA), but rather we defined gates based on the 2D plot of moxGFP versus mCherry. This is explicitly marked in Figure 3A.

On page 9, the authors claim that their data supports a model that rs4633 increases RNA abundance, leading to higher COMT expression. Can the authors rule out a model whereby rs4633 facilitates translation initiation, as suggested by Tsao et al. 2011, leading to both an increase in mRNA and protein abundance?

Response: Thanks for this question and opportunity to clarify. We have now added a sentence to the Discussion and included the following paragraph in the Supplementary Note:

“Importantly, our study does not rule out a model where rs4633 facilitates translation initiation. Nevertheless, our data suggest a potential concurrent mechanism where rs4633 leads to higher protein abundance in human cell lines and in an *in vitro* translation assay (Tsao et al. 2011) by increasing RNA abundance. We note that Tsao et al did not directly measure RNA abundance in their study. In Supplementary Figure 3A of Nackley et al 2006, the APS haplotype containing rs4633 C>T showed slightly higher total RNA abundance compared to the LPS haplotype (in our study, the wild-type template). However, this was not statistically significant and was only observed for the S-COMT isoform. It is possible that our observations are compatible with the conclusions in Tsao et al. 2011. For example, increased translation of rs4633 C>T may lead to stabilization of the RNA.”

Full Revision

The paper references "effect size" at multiple points (e.g. "polysome effect size") but we could not find this term explicitly defined (for example: for the polysome effect size, were RNA counts for each polysome fraction divided by the relative abundance of that RNA in total RNA?)

Response: We apologize for this confusion. Please see our response above. We have also stated the definition of effect size explicitly in the revised manuscript.

Could you elaborate on how you define "protein abundance and "effect size: in Figure 5G? How is enrichment in P3 or P1 calculated?

Response: Effect size is defined as described above. Enrichment in P3 or P1 is calculated with respect to the abundance in gDNA (unsorted cells).

Were 3396 variants considered for all readouts in this paper? How many of these variants were present in each ROI? It may be worth clarifying sample sizes.

Response: Thanks for the suggestion. The reviewer is correct: 3396 variants were present in all biological replicates and all readouts (after excluding polysome metafractions 1 and 2 and flow cytometry population 4). The Methods were updated to include all readouts that were dropped. The number of variants in each ROI are now included in this section of the main text.

How did Twist generate these mutagenized sequences? We assumed that they used error-prone PCR due to the mention of multiple nucleotide polymorphisms, but couldn't find an explicit answer.

Response: Twist generates these mutagenized inserts using degenerate primers. This allows all alternate codons to be assayed (all silent, missense changes). This is now noted in the Methods.

<https://www.twistbioscience.com/resources/technical-note/solid-phase-dna-synthesis-allows-tight-control-combinatorial-library>

In the methods, it may be worth elaborating on the composition of the HsCD00617865 plasmid. For example: this COMT reporter is under the control of a constitutively-expressed T7 promoter, correct?

Response: The HsCD00617865 plasmid was only used as a template for PCR amplification and generation of the transgene. The transgene is cloned into a vector containing attB sites for recombination into the landing pad cell line (Matreyek et al 2020). Transcription is induced by Doxycycline from the landing pad locus. Plasmid maps used for transfection into the landing pad line are now included in the GitHub repository.

In Supplementary Figures 4 and 5, it would be helpful to explicitly say that you are reporting Pearson correlations between biological replicates.

Response: Thanks for the suggestion. The legends have been updated accordingly.

"After summarizing biological replicates (N=4) for each readout...": how did the authors summarize biological replicates? Were counts averaged?

Response: Biological replicates were summarized using the median. This is now clarified in the Methods.

The authors used pairwise correlations between flow cytometry fractions, polysome fractions, and total RNA/gDNA as indications of data quality. Do the authors expect for these counts to be strongly correlated? We would not necessarily expect to see a strong correlation between ribosome load and RNA/gDNA.

Response: We used replicate correlation as an indicator of data quality. Our readouts of ribosome load reflect the abundance of a variant in a particular polysome fraction. Given that variants that are highly abundant in the RNA pool will on average be more highly represented in polysome fractions, we would expect a correlation between the abundance of a variant in total RNA and in polysome fractions.

The authors may need to check that their standard deviations on fold changes are properly reported.

Response: In the Figures and the main text, we specified the confidence intervals as calculated by ALDEx2 method instead of reporting standard deviations on fold changes. Specifically, the confidence intervals were determined by Monte Carlo methods that produce a posterior probability distribution of the observed data given repeated sampling. Variants in which the confidence intervals do not cross 0 are considered true discoveries (section 5.4.1 of the ALDEx2 vignette on Bioconductor).
https://www.bioconductor.org/packages/devel/bioc/vignettes/ALDEx2/inst/doc/ALDEx2_vignette.html#541_The_effect_confidence_interval

We would expect standard deviation bounds to be symmetric for log fold changes, but not on unlogged fold changes - for example see page 8, for the sentence "our point estimate for nonsense variant effects on COMT RNA abundance was approximately a two-fold decrease relative to the gDNA frequency (fold change of 0.43 +/- 0.13; mean +/- standard deviation; Methods)."

Response: Thanks for the suggestion. To avoid any confusion about the symmetry, we replaced the +/- notation, and explicitly noted the mean and standard deviation. To help the reader gain an intuition of the magnitude of variant effects, we conducted a frequency based normalization-dependent analysis using limma (as previously employed in Hoskins et al. 2023. Genome Biology). We now report a fold change (unlogged) for RNA abundance compared to gDNA abundance. The point estimate is the mean and s.d. across all nonsense variants.

On page 10, the authors say that their data suggests that hydrophobicity in the early coding region of COMT may be important for COMT folding. If this is the case, would we expect to see this effect in flow cytometry data (which is affected by protein degradation) and not polysome profiling (which is unaffected by post-translational protein degradation)?

Response: We apologize as we are uncertain about the reviewer's intended question. The section that refers to the importance of hydrophobicity indeed refers to the flow cytometry data.

Full Revision

While there are specific instances in which the amino acid properties encoded by the mRNA influences translation dynamics, these are not universally true. Consequently, we did not expect these impacts to be observed at the level of polysome profiling.

We believe that we would have some trouble replicating the analysis from this paper from the raw data, given that the bulk of the analysis on GitHub is presented as a single R Markdown file, with references to local files to which we do not have access. We recommend that the authors add additional documentation to their repository to facilitate re-analysis.

Response: Thanks for the opportunity to address this issue of critical importance. To facilitate replication, we have now deposited all analysis files to Zenodo and refactored the code to enable replication by simply running a markdown file.

In Figure 1B, indicating that more signal indicates less structure (in the legend or the figure itself) may assist readers who are unfamiliar with DMS-seq.

Response: Thanks for the suggestion. This is now updated.

Figure 1C does a great job presenting evidence for the translation of uORFs, but does not seem to flow with the overall argument of the paper, so may fit better in the supplement.

Response: We considered this suggestion, and opted for keeping its placement as it gives evidence that our transgene is translated primarily as the MB-COMT isoform. This ensures that, for variants upstream of the S-COMT isoform, we can assay effects on ribosome load that are tied to mechanisms of translation elongation and codon stability.

We believe there is a typo in the Figure 1 legend that should read "K562" instead of "H562".

Response: Thank you, this was indeed a typo.

You also gated to separate into P1-P4, correct? Can you also show the bounds of that gating strategy in Figure 3A?

Response: This has been updated. We also added the gating strategy in response to comments from reviewer #1.

We find Figure 3F very compelling. Do you have any theories as to why mutating I59-H66 to nonpolar, uncharged residues leads to increased COMT expression?

Response: We do not have any theories for why this may be. However, we noted that with the exception of V63, residues I59-H66 are not evolutionarily constrained (based on DeMask entropy values). This suggests mutational tolerance for nonpolar, uncharged residues in this region (with the exception of V63 and H66; see Figure 3D).

There appears to be a non-negligible proportion of di- and tri- nucleotide polymorphisms in Supplementary Figure 4. Were these excluded in downstream analyses?

Full Revision

Response: These variants are expected from the Twist mutagenesis strategy and included in analysis. We believe they are at lower frequency compared to SNPs due to less favorable annealing of the degenerate primers.

A minor typo in the discussion reads "fluoresce".

Response: Done

Significance

Describe the nature and significance of the advance (e.g. conceptual, technical, clinical) for the field.

This work investigated the regulatory effects of thousands of coding variants in the COMT gene, focusing on two regions with clinical significance, by using high-throughput reporter assays. The results from this will be useful for clinical scientists interested in understanding the impacts of COMT mutations and be a useful framework for other systems/computational biologists to understand the impacts of coding mutations across different levels of regulatory function. Mutations in protein regions, if having a function, are classically known to interfere with protein function. There are fewer large-scale efforts to understand the impacts of coding mutations affecting expression through potentially changing of RNA structure or codon optimization - this work has contributed towards that frontier.

Place the work in the context of the existing literature (provide references, where appropriate). This is (as far as I am aware) the first paper that has integrated high-throughput screens massively parallel reporter assays from RNA degradation, ribosomal load, and flow cytometry. Previous papers have tended to measure on expression regulation on only one dimension (i.e. Greisemer et al. 2023 on RNA degradation, Sample et al. 2019 on ribosomal load, and de Boer et al. 2020 on protein expression).

Response: Thanks for highlighting the novelty of our approach compared to existing strategies in the literature.

State what audience might be interested in and influenced by the reported findings.

Clinicians/researchers interested in COMT, computational biologists, geneticists and potentially structural biologists interested in understanding the consequences of amino acid mutations on RNA/protein expression

Response: Thanks for noting the broad significance of our study.

Define your field of expertise with a few keywords to help the authors contextualize your point of view. Indicate if there are any parts of the paper that you do not have sufficient expertise to evaluate.

Genomics, Massively parallel reporter assays, High-throughput regulatory screens.

Reviewer #3 (Evidence, reproducibility and clarity (Required)):

This manuscript reports on transcript sequence variants that affect expression of the gene COMT. Targeted analysis of SNPs identifies 5' UTR variants that affect COMT, leading to the identification of translated uORFs. Common coding sequence SNPs do not affect COMT expression, however. Massively parallel analyses of mRNA abundance, protein abundance, and translation are combined to look more broadly at coding sequence variants. These analyses focus on regions of predicted structure in the COMT transcript. Both silent and missense mutations that increase mRNA abundance are identified. Protein abundance is then measured and many missense mutations are found to change protein levels. To address translation directly, analysis of polysome loading is performed and significant differences are identified, although technical challenges limit data quality in these experiments. These different experiments are then analyzed jointly to classify mutation effects and identify a class of silent mutations with expression effects, leading to a proposal that these act through structure.

The joint, integrative analysis of COMT variants through a range of methods allows clearer insights into interconnected post-transcriptional effects. The massively parallel experiments generate high-quality data, although targeted validation of key results would strengthen the work. The findings advance our understanding of silent variant effects, which remains an open question, and technical innovations could find broader applications.

Response: Thanks for the positive assessment of the quality of the data generated and the potential for the broader application of the technical innovations.

I do have concerns with the present version of this work.

1. There is no validation presented for high-throughput experimental data. I would say that validating the effects of M152T and V63V variants from Figure 2B would substantially strengthen the work and support key conclusions.

Response: Our experiments collectively enabled nearly 10,000 measurements of variant effect (summed over three layers of gene expression). The goal of our study was to identify broad mechanisms of variant effect. While we are excited about the specific variants uncovered, targeted experimental methods for validating changes to RNA abundance, such as RT-qPCR, are unlikely to be sufficiently sensitive. For example, RNA abundance effects in our study had a median effect size of 1.47 for variants up in RNA, and 0.4 for variants down in RNA. This likely corresponds to less than one Ct difference between the variant and the reference allele. Indeed, previous studies such as Findlay et al., 2018 *Nature* that reported similar effect sizes (< two-fold) for variants in BRCA1 on RNA expression did not report any targeted validation experiments. The challenge of validating small changes is also evident in a recent study by Lim et al., 2021, in which 6 and 9 biological replicates were used to reach marginally significant p-values for RNA abundance effects for 5' UTR variants in *FGF7* and *FOS*, respectively (Figure 4B).

Thus, for time and cost concerns, we respectfully suggest that targeted experiments involving V63V and M152T are beyond the scope of our study. Nevertheless, to further strengthen our conclusions, we have computationally confirmed our findings using a different analysis framework. We found 75/76 of the variants significant by ALDEx2 analysis were also significant by limma analysis (a frequency based normalization-dependent analysis, as previously employed in Hoskins et al. 2023. *Genome Biology*) using the same FDR (0.1).

2. In the fluorescent reporter scheme, it seems that variants reducing mRNA abundance should be enriched in the "P2" gate region relative to "P1", as they would have lower mRNA abundance and correspondingly lower protein abundance. However, this analysis is not performed, and instead P1 and P3 are compared (Figure 3G), which would seem to focus on protein-level effects.

Response: Our initial hesitation in comparing P2 to P1 is that the P2 population may be enriched for cells that underwent inefficient induction of transcription with Doxycycline. Hence technical factors as opposed to the effect of the variants may dominate this comparison. In response to the reviewer's comments, we carried out the suggested analysis (new Supplementary Figure 5B). We found that variants that are down in RNA are enriched in P2 relative to P1 as expected. This is now noted in the Results section.

3. In general the work classifies variants in several different ways and it would help to be a little clearer in naming these classes. For instance, in describing the FACS-based analysis of variant expression it is written, "protein fluorescence conditioned on RNA fluorescence" which is confusing at best-it's a fluorescence-based measurement that is used indirectly to measure COMT reporter abundance.

Response: Thanks for the suggestion. We agree that our initial word-choice was imprecise. We rewrote this section to indicate mCherry fluorescence is an indirect proxy for RNA abundance.

4. Likewise, the populations with shifted GFP/mCherry ratio in this assay are described as "uncorrelated" populations, which is opaque and somewhat inaccurate-there seems to be a correlation in this group but at a different ratio.

Response: We have revised the language in the manuscript. We opted for "low or high RNA/protein abundance" to indicate the relationship between GFP and mCherry fluorescence in populations P3 and P4.

5. In the same way, "deleterious variants" is used to describe protein abundance changes, but this term implies a fitness effect and is not very specific.

Response: We apologize for the confusing word choice. We did away with this term in favor of "variants with low protein abundance".

6. In discussing the effects of missense COMT variants on protein levels, there is an implicit assumption that degradation of mis-folded protein (or perhaps properly-folded protein with excess hydrophobic exposure?) explains these effects. This is plausible, but it would help to lay out this reasoning more clearly.

Response: Thanks for the suggestion. We have added a sentence at the end of the section that specifies this assumption and cites a recent study reporting that rare missense variants in COMT may be misfolded and degraded by the proteasome (Larsen et al. 2023).

7. It is written that, "In line with codon stability as a predictor of translational efficiency (Presnyak et al., 2015), variants with low codon optimality were depleted from polysomes compared to variants with optimal codons". However, this mis-states the conclusions of the cited study, which

notes, "Importantly, under normal conditions the ribosome occupancy of the HIS3 opt and non-opt constructs was determined to be similar (Fig. 6B)".

Response: We apologize for mis-stating the conclusions of Presnyak et al. 2015. We have now revisited the relevant literature to more accurately place our conclusions in the context of literature. While Presnyak et al. and several other studies (Bazzini et al., 2016; Mauger et al., 2019) have clearly linked the association between codon choice and mRNA stability. We now reference Mauger et al. 2019 who used elegant experiments to demonstrate that mRNA secondary structure is a driver of increased protein production and synergizes with codon optimality (Figure 5B). Their results further support the role of codon optimality on RNA stability while providing evidence of additive impact on translation efficiency.

8. It is written that, "One intriguing possibility is to develop multiplexed assays of variant effect on RNA folding, using mutational profiling RNA probing methods (Weng et al., 2020; Zubradt et al., 2017)." How would this differ from the "Mutate and Map" approach in doi://10.1038/nchem.1176 and subsequent work from the same group?

Response: Thanks for pointing out the more recent work following the initial papers in 2010-2011. We have missed the work from the Das lab that extended the Mutate and Map approach to utilize mutational profiling (Cheng and Kladwang et al., 2017). We updated our Discussion to indicate that the proposed assay has been pioneered and is a viable approach for high-throughput determination of variant effects on RNA folding.

Because mutational profiling methods leverage reverse transcriptase readthrough and mismatch incorporation, they enable deeper and more uniform coverage of sequencing reads, particularly for longer transcripts. A key design principle of the proposed assay is to mutagenize only certain types of variants in the library such that they do not overlap RT mismatch signatures arising from the RNA probing reagent/RT enzyme. For example, readthrough of DMS base adducts largely generates A>N or C>N mismatches, so a variant library would be designed to only contain variants at G or T bases. This ensures variants in the library can be differentiated from signals of the RNA probing method.

Referees cross-commenting

I generally agree with the other reviewers and found that many small points on the figures were confusing, and in some cases the values being computed and displayed were under-specified.

I agree with Reviewer 1 that the polysome fractionation probably has limited power due to experimental design, and that the interpretation of changed ribosome loading is subtle.

Response: In response to these helpful comments, we have clarified the points highlighted by the reviewers and expanded the limitations section related to the ribosome loading assay. Thanks for these constructive suggestions to strengthen our study.

Reviewer #3 (Significance (Required)):

The joint, integrative analysis of COMT variants through a range of methods allows clearer insights into interconnected post-transcriptional effects. The massively parallel experiments generate high-quality data, although targeted validation of key results would strengthen the

Full Revision

work. The findings advance our understanding of silent variant effects, which remains an open question, and technical innovations could find broader applications.

Response: Thanks for pointing out the high-quality of the generated data and the broad significance of our study. The goal of our study was to identify broad mechanisms of variant effect instead of focusing on differential expression for any specific variants.

27th Nov 2023

Manuscript Number: MSB-2023-12096

Title: Integrated multiplexed assays of variant effect reveal cis-regulatory determinants of catechol-O-methyltransferase gene expression

Dear Prof. Cenik,

Thank you again for submitting your revised study to Molecular Systems Biology along with the referee reports from Review Commons. We have now heard back from the two reviewers who agreed to evaluate your revised study. We invited the same reviewers who reviewed the manuscript at Review Commons. (Please note that reviewer #2 is the previous reviewer #3 from Review Commons.) As you will see below, the reviewers think that the study has improved as a result of the performed revisions. However, reviewer #1 still lists a few concerns, which we would ask you to address in a revision.

We would also ask you to address the following editorial points:

- Please provide a .doc version of the manuscript text (including legends for the main figures) and individual production quality figure files for the main Figures (one file per figure).
- Please note that our editorial policy does not allow "data not shown" statements. All discussed data should be included either in the main figures, Appendix or Datasets.
- We typically do not allow subheadings in the Discussion, please remove the subheading "Limitations".
- Supplementary Tables 1-4 should be provided as Datasets EV1-EV4. Please provide one file per dataset and include in each .xls file a separate tab with the description of the dataset.
- We have replaced Supplementary Information by the Expanded View (EV format). In this case, all additional figures can be included in a PDF called Appendix. Appendix Figures should be labeled and called out as: "Appendix Figure S1, Appendix Figure S2, ... etc.". Each Appendix Figure legend should be provided below the corresponding Figure in the Appendix. Please include a Table of Contents in the beginning of the Appendix. For detailed instructions regarding expanded view please refer to our Author Guidelines: .
- The Supplementary Note should be renamed "Appendix Note" and included in the Appendix.
- All Materials and Methods need to be described in the main text. We would encourage you to use 'Structured Methods', our new Materials and Methods format. According to this format, the Material and Methods section should include a Reagents and Tools Table (listing key reagents, experimental models, software and relevant equipment and including their sources and relevant identifiers) followed by a Methods and Protocols section in which we encourage the authors to describe their methods using a step-by-step protocol format with bullet points, to facilitate the adoption of the methodologies across labs. More information on how to adhere to this format as well as downloadable templates (.doc or .xls) for the Reagents and Tools Table can be found in our author guidelines: . An example of a Method paper with Structured Methods can be found here:
- Please include a Data availability section describing how the data and code have been made available. This section needs to be formatted according to the example below:
The datasets and computer code produced in this study are available in the following databases:
 - Chip-Seq data: Gene Expression Omnibus GSE46748 (<https://www.ncbi.nlm.nih.gov/geo/query/acc.cgi?acc=GSE46748>)
 - Modeling computer scripts: GitHub (<https://github.com/SysBioChalmers/GECKO/releases/tag/v1.0>)
 - [data type]: [full name of the resource] [accession number/identifier] ([doi or URL or identifiers.org/DATABASE:ACCESSION])
- Please include a "Disclosure Statement & Competing Interests" in the main text.
- Please provide a "standfirst text" summarizing the study in one or two sentences (approximately 250 characters), three to four "bullet points" highlighting the main findings and a "synopsis image" (550px width and max 400px height, jpeg format) to highlight the paper on our homepage.
- For data quantification: please specify the name of the statistical test used to generate error bars and P values, the number (n) of independent experiments (specify technical or biological replicates) underlying each data point and the test used to calculate p-values in each figure legend. The figure legends should contain a basic description of n, P and the test applied. Graphs must include a description of the bars and the error bars (s.d., s.e.m.).

- When you resubmit your manuscript, please download our CHECKLIST (<https://bit.ly/EMBOPressAuthorChecklist>) and include the completed form in your submission.

Please note that the Author Checklist will be published alongside the paper as part of the transparent process (<https://www.embopress.org/page/journal/17444292/authorguide#transparentprocess>).

Please resubmit your revised manuscript online, with a covering letter listing amendments and responses to each point raised by the referees. Please resubmit the paper ****within one month**** and ideally as soon as possible. If we do not receive the revised manuscript within this time period, the file might be closed and any subsequent resubmission would be treated as a new manuscript. Please use the Manuscript Number (above) in all correspondence.

Click on the link below to submit your revised paper.

Link Not Available

Yours sincerely,

Maria Polychronidou

Maria Polychronidou, PhD
Senior Editor
Molecular Systems Biology

If you do choose to resubmit, please click on the link below to submit the revision online before 27th Dec 2023.

Link Not Available

IMPORTANT: When you send your revision, we will require the following items:

1. the manuscript text in LaTeX, RTF or MS Word format
2. a letter with a detailed description of the changes made in response to the referees. Please specify clearly the exact places in the text (pages and paragraphs) where each change has been made in response to each specific comment given
3. three to four 'bullet points' highlighting the main findings of your study
4. a 'standfirst text' summarizing in two sentences the study (approx. 250 characters)
6. a "thumbnail image" (width=211 x height=157 pixels, jpeg format), which can be used as 'visual title' to highlight your paper on our homepage.
7. Please include an author contributions statement after the Acknowledgements section (see <https://www.nature.com/msb/authors/index.html#Submission>)
8. When assembling figures, please refer to our figure preparation guideline in order to ensure proper formatting and readability in print as well as on screen:
<https://bit.ly/EMBOPressFigurePreparationGuideline>
See also figure legend guidelines: <https://www.embopress.org/page/journal/17444292/authorguide#figureformat>

*** PLEASE NOTE *** As part of the EMBO Publications transparent editorial process initiative (see our Editorial at <https://www.nature.com/msb/journal/v6/n1/full/msb201072.html>), Molecular Systems Biology will publish online a Review Process File to accompany accepted manuscripts. When preparing your letter of response, please be aware that in the event of acceptance, your cover letter/point-by-point document will be included as part of this File, which will be available to the scientific community. More information about this initiative is available in our Instructions to Authors. If you have any questions about this initiative, please contact the editorial office (msb@embo.org).

Reviewer #1:

In this manuscript, the authors used multiple Multiplexed Assays of Variant Effects (MAVEs) to measure the impact of variation in 5' UTR and coding sequences of expression of the COMT gene. COMT encodes an enzyme that catabolizes many neurotransmitters and COMT variants have been linked to a number of human psychiatric conditions. The authors assayed RNA levels, polysome loading, and protein levels resulting from COMT variants. They observed a decrease in RNA levels for nonsense mutation variants (as expected). Importantly, they also report a variety of expression effects that result from missense

and silent mutations. Overall, the work offers a rigorous and comprehensive assessment of the impact of sequence variation on molecular expression, with the caveat that biochemical function was not assayed. This is an important study that should appeal to both basic researchers interested in MAVE and MPRA experimental modalities and neurobiologists and clinicians interested in disorders involving COMT activity.

This revision is a substantial improvement over an earlier preprint of the manuscript. For some reason, the changes were not tracked, making it difficult to see which changes were made. There are still some issues that the authors should address, numbered as follows:

- 1) Is it possible that the variants that affect RNA levels might either remove or add transcription factor binding sites? Perhaps the authors could search for TFBS for each wildtype / variant pair that affect RNA level, or at least discuss this as a possibility in the limitations section.
- 2) The sentence, "We note that our design is unable to determine if protein abundance effects reflect underlying effects on RNA abundance" is confusing. I think it was added as a response to the preprint review, but it is. However, the design of the assay assumes that mCherry is a proxy for measuring RNA levels, which is reasonable. Please revise this for clarity.
- 3) Related to point 2, the authors found that nonsense variants were enriched in the "low protein" population (P3). This population has low GFP and low mCherry, so it's really a "low-protein / low-RNA" population. This fits with the interpretation that these variants largely lead to NMD, decreasing both GFP and mCherry. Later on, the authors found these variants were depleted from deeper polysome fractions and attribute that to NMD, "...removal of these mRNAs from the translating pool" and "We propose a modest effect for nonsense variants in the mid-polysome..." However, the interpretation that this is due to NMD does not make sense. The reasons why nonsense variants are depleted in the polysome are probably 1) they are generally depleted in RNA level, so they are depleted from total RNA and polysomal fractions (not a translational effect) and 2) because they make the COMT-GFP ORF much shorter so the reporters can't physically accommodate as many ribosomes. The authors address this later in the discussion, but it should be clarified in the main results.
- 4) The paragraph starting "The reliability of polysome profiling in reporting active translation is debated..." describes a lack of correlation between ribosome load and protein abundance, including a discussion of NMD. A relevant paper came out earlier this year (PMID: 37227054) that assayed RNA levels, protein levels, and polysome loads for variations of yeast 5' UTRs. It might make sense to discuss it here, as it reported a fairly strong correlation in effects on protein expression and polysome load.
- 5) The authors cite Li and Chen 2023, but it doesn't appear to be in the reference list.
- 6) The methods have an underlined section "We only considered variants ... median log-frequency". Is that log base 10? Please specify.
- 7) A minor suggestion: I found it a little difficult to follow "region 1" and "region 2" in the text. You might want to call them "ROI1" and "ROI2" or Exon3 and Exon4 variants.
- 8) The sentence, "For each variant, normalized frequencies in the negative control were subtracted..." is confusing to read. Do you mean perhaps, "Variants that had normalized frequencies lower than the negative control were removed from further analyses"?

Reviewer #2:

Revisions have addressed my concerns with the original submission and I support its publication in the present form.

Rev_Com_number: RC-2023-02121

New_manu_number: MSB-2023-12096

Corr_author: Cenik

Title: Integrated multiplexed assays of variant effect reveal cis-regulatory determinants of catechol-O-methyltransferase gene expression

Reviewer #1:

In this manuscript, the authors used multiple Multiplexed Assays of Variant Effects (MAVEs) to measure the impact of variation in 5' UTR and coding sequences of expression of the COMT gene. COMT encodes an enzyme that catabolizes many neurotransmitters and COMT variants have been linked to a number of human psychiatric conditions. The authors assayed RNA levels, polysome loading, and protein levels resulting from COMT variants. They observed a decrease in RNA levels for nonsense mutation variants (as expected). Importantly, they also report a variety of expression effects that result from missense and silent mutations. Overall, the work offers a rigorous and comprehensive assessment of the impact of sequence variation on molecular expression, with the caveat that biochemical function was not assayed. This is an important study that should appeal to both basic researchers interested in MAVE and MPRA experimental modalities and neurobiologists and clinicians interested in disorders involving COMT activity.

This revision is a substantial improvement over an earlier preprint of the manuscript. For some reason, the changes were not tracked, making it difficult to see which changes were made. There are still some issues that the authors should address, numbered as follows:

Response: We thank the reviewer for their constructive suggestions that substantially strengthen our manuscript. We address all the remaining issues.

1) Is it possible that the variants that affect RNA levels might either remove or add transcription factor binding sites? Perhaps the authors could search for TFBS for each wildtype / variant pair that affect RNA level, or at least discuss this as a possibility in the limitations section.

Response: Thanks for the suggestion. For variants significantly altering RNA abundance, we searched the wild-type and variant sequences for TF binding sites and included an additional table reporting these results (Dataset EV1). The results of this analysis are now included on page 7.

2) The sentence, "We note that our design is unable to determine if protein abundance effects reflect underlying effects on RNA abundance" is confusing. I think it was added as a response to the preprint review, but it is. However, the design of the assay assumes that mCherry is a proxy for measuring RNA levels, which is reasonable. Please revise this for clarity.

Response: We agree with the reviewer that this sentence was confusing. We added the word "unequivocally" to this sentence to clarify this caveat (page 8).

3) Related to point 2, the authors found that nonsense variants were enriched in the "low protein" population (P3). This population has low GFP and low mCherry, so it's really a "low-protein / low-RNA" population. This fits with the interpretation that these variants largely lead to NMD, decreasing both GFP and mCherry. Later on, the authors found these variants were depleted from deeper polysome fractions and attribute that to NMD, "...removal of these mRNAs from the translating pool" and "We propose a modest effect for nonsense variants in the mid-polysome..." However, the interpretation that this is due to NMD does not make sense. The reasons why nonsense variants are depleted in the polysome are probably 1) they are generally depleted in RNA level, so they are depleted from total RNA and polysomal fractions (not a translational effect) and 2) because they make the COMT-GFP ORF much shorter so the reporters can't physically accommodate as many ribosomes. The authors address this later in the discussion, but it should be clarified in the main results.

Response: We agree that the most parsimonious explanation for our observation is that the COMT ORF is shorter, and we revised our Results section to indicate this as the primary interpretation (page 10).

4) *The paragraph starting "The reliability of polysome profiling in reporting active translation is debated..." describes a lack of correlation between ribosome load and protein abundance, including a discussion of NMD. A relevant paper came out earlier this year (PMID: 37227054) that assayed RNA levels, protein levels, and polysome loads for variations of yeast 5' UTRs. It might make sense to discuss it here, as it reported a fairly strong correlation in effects on protein expression and polysome load.*

Response: Thanks for highlighting this critical study. We have included a mention of this in the Discussion (page 13) and added a new section in the Appendix Note discussing potential reasons for the discrepancy between the two studies.

5) *The authors cite Li and Chen 2023, but it doesn't appear to be in the reference list.*

Response: We confirmed this citation is in the References section.

6) *The methods have an underlined section "We only considered variants ... median log-frequency". Is that log base 10? Please specify.*

Response: We clarified that this is a natural log in the Methods (page 28).

7) *A minor suggestion: I found it a little difficult to follow "region 1" and "region 2" in the text. You might want to call them "ROI1" and "ROI2" or Exon3 and Exon4 variants.*

Response: Thanks for the suggestion. We revised the text and consistently used the "ROI 1 and ROI 2" nomenclature.

8) *The sentence, "For each variant, normalized frequencies in the negative control were subtracted..." is confusing to read. Do you mean perhaps, "Variants that had normalized frequencies lower than the negative control were removed from further analyses"?*

Response: We reworded this sentence as suggested (Methods, page 27).

3rd Jan 2024

Manuscript Number: MSB-2023-12096R

Title: Integrated multiplexed assays of variant effect reveal determinants of catechol-O-methyltransferase gene expression

Dear Dr. Cenik,

I hope you had a good start in the New Year! Thank you for sending us your revised manuscript, we have now gone through the revisions and we think that all remaining issues of the referees have now been addressed. There are however a few minor editorial points that we would ask you to fix before acceptance:

- Our data editors have noted the following regarding the figure legends:

- The legends for figures 1a-e are incorrectly labelled as 1b-e in the manuscript. This needs to be corrected.
- Please indicate the statistical test used for data analysis in the legends of figure 4a; EV 2b-c.
- The box plots need to be defined in terms of minima, maxima, and whiskers, in the legends of figures 2c; 3g; EV 5b.
- The box plots need to be defined in terms of minima, maxima, centre, bounds of box and whiskers, and percentile in the legends of figures 4d; 5g; EV 2b; EV 5c.
- Please include the information related to n in the legends of figures 2c-d; 3c, g; 4a-b, d; EV 2b-c; EV 5b-c.
- The measure of center for the error bars needs to be defined in the legends of figures 2d; 3c; 4b.

- Please remove the 'Authors Contributions' from the manuscript. The 'Author Contributions' section is replaced by the CRediT contributor roles taxonomy to specify the contributions of each author in the journal submission system. Please use the free text box in the 'author information' section of the online submission system to provide more detailed descriptions if needed (e.g., 'X provided intracellular Ca⁺⁺ measurements in fig Y').

- Please note that our editorial policy does not allow "data not shown". Please provide the information regarding the outliers mentioned in the figure legend for figure 2C as not shown (page 42).

- The descriptions of the Datasets should be removed from the text file.

- In the Appendix: please update the document title (on the first page) to Appendix for "Integrated multiplexed assays of variant effect reveal determinants of catechol-O-methyltransferase gene expression".

- The Source Data checklist needs to be completed.

- Please reorganize the Source data files to one zip folder per figure (ie. one zip folder for Figure 3 and one zip folder for Figure 4).

Please resubmit your revised manuscript online ****within one month**** and ideally as soon as possible. If we do not receive the revised manuscript within this time period, the file might be closed and any subsequent resubmission would be treated as a new manuscript. Please use the Manuscript Number (above) in all correspondence.

Click on the link below to submit your revised paper.

Link Not Available

Kind regards,

Maria

Maria Polychronidou, PhD
Senior Editor
Molecular Systems Biology

If you do choose to resubmit, please click on the link below to submit the revision online before 2nd Feb 2024.

Link Not Available

*** PLEASE NOTE *** As part of the EMBO Publications transparent editorial process initiative (see our Editorial at <https://www.nature.com/msb/journal/v6/n1/full/msb201072.html>), Molecular Systems Biology will publish online a Review Process File to accompany accepted manuscripts. When preparing your letter of response, please be aware that in the event of acceptance, your cover letter/point-by-point document will be included as part of this File, which will be available to the scientific community. More information about this initiative is available in our Instructions to Authors. If you have any questions about this initiative, please contact the editorial office (msb@embo.org).

Rev_Com_number: RC-2023-02121

New_manu_number: MSB-2023-12096R

Corr_author: Cenik

Title: Integrated multiplexed assays of variant effect reveal determinants of catechol-O-methyltransferase gene expression

All editorial and formatting issues were resolved by the authors.

18th Jan 2024

Manuscript number: MSB-2023-12096RR

Title: Integrated multiplexed assays of variant effect reveal determinants of catechol-O-methyltransferase gene expression

Dear Dr. Cenik,

Thank you again for sending us your revised manuscript. We are now satisfied with the modifications made and I am pleased to inform you that your paper has been accepted for publication.

Kind regards,

Maria

Maria Polychronidou, PhD
Senior Editor
Molecular Systems Biology

Rev_Com_number: RC-2023-02121

New_manu_number: MSB-2023-12096RR

Corr_author: Cenik

Title: Integrated multiplexed assays of variant effect reveal determinants of catechol-O-methyltransferase gene expression